# Spiking Neural Network as Adaptive Event Stream Slicer

**Jiahang Cao[1]\* Mingyuan Sun[2]\* Ziqing Wang[3]\* Hao Cheng[1]**
**Qiang Zhang[1,4] Shibo Zhou[5] Renjing Xu[1]**
[1] The Hong Kong University of Science and Technology (Guangzhou)
[2] Northeastern University [3] Northwestern University
[4] Beijing Innovation Center of Humanoid Robotics Co. Ltd. [5] Brain Mind Innovation
{jcao248, hcheng046, qzhang749}@connect.hkust-gz.edu.cn
mingyuansun20@gmail.com, ziqingwang2029@u.northwestern.edu
bob@brain-mind.com.cn, renjingxu@hkust-gz.edu.cn

## Abstract

Event-based cameras are attracting significant interest as they provide rich edge information, high dynamic range, and high temporal resolution. Many state-of-the-art event-based algorithms rely on splitting the events into fixed groups, resulting in the omission of crucial temporal information, particularly when dealing with diverse motion scenarios (*e.g.*, high/low speed). In this work, we propose **SpikeSlicer**, a novel-designed plug-and-play event processing method capable of splitting events stream adaptively. SpikeSlicer utilizes a low-energy spiking neural network (SNN) to trigger event slicing. To guide the SNN to fire spikes at optimal time steps, we propose the Spiking Position-aware Loss (SPA-Loss) to modulate the neuron's state. Additionally, we develop a Feedback-Update training strategy that refines the slicing decisions using feedback from the downstream artificial neural network (ANN). Extensive experiments demonstrate that our method yields significant performance improvements in event-based object tracking and recognition. Notably, SpikeSlicer provides a brand-new SNN-ANN cooperation paradigm, where the SNN acts as an efficient, low-energy data processor to assist the ANN in improving downstream performance, injecting new perspectives and potential avenues of exploration. Our code is available at https://github.com/AndyCao1125/SpikeSlicer.

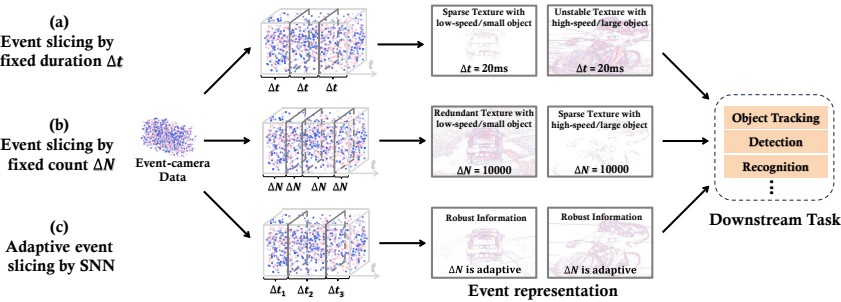

Figure 1: Comparison of event slicing methods. Traditional methods slice event streams based on prefixed time intervals (a) or event counts (b). In contrast, our approach (c) utilizes SNN as a dynamic event processor for adaptive event slicing. The sliced sub-event streams can be converted into various event representations with robust information and then applied to multiple downstream tasks.

---

\*Equal contribution.

38th Conference on Neural Information Processing Systems (NeurIPS 2024).

# 1 Introduction

Event-based cameras [1] are bio-inspired sensors that capture event streams in an asynchronous and sparse way. Compared with conventional frame-based cameras, event-based cameras offer numerous outstanding properties: high temporal resolution (with the order of $\mu s$), high dynamic range (higher than 120 dB), low latency, and low power consumption. Over recent years, rapid growth has been witnessed in dealing with event data due to the inherent advantages of event-based cameras, such as object tracking [2, 3], depth estimation [4, 5], and recognition [6, 7]. Before applying to various downstream tasks, the event stream must be split by groups and then transformed into different event representations, *e.g.*, frame, voxel, or point for deep learning architectures.

In detail, the process of event-to-representation conversion consists of two steps: (1) slicing the raw event stream into multiple sub-event stream groups, and (2) converting these sub-event streams into different event representations. Much of the current research focuses on the second step, aiming to refine event representation [8] techniques such as time surface [6] and event spike tensor [9]. Yet, this focus often overlooks the crucial first step of slicing, where issues such as the non-uniformity of the information contained in the fixed-sliced event remain unaddressed.

To address the challenges in the slicing process, we delve into the limitations of traditional slicing techniques. Common methods typically cut the event stream into several fixed groups. For example, slicing event stream with fixed event count [10] or fixed time intervals [11, 12] as depicted in Fig. 1. However, these fixed-group slicing techniques often lead to problems: they may cause insufficient information capture in low-speed motion scenarios or excessive redundancy in high-speed conditions, thereby failing to accurately capture the dynamic changes in event distribution. Additionally, some hyper-parameters, *e.g.*, the length of time interval, are highly-sensitive to the downstream tasks (examples are provided in Appendix C) and must be carefully pre-determined. Although some latest slicing methods [13, 14] propose to adaptively sample the events, there still exists the problem of hyper-parameter tuning which can not achieve a fully learnable and adaptable slicing process.

In order to address the above issues, we propose SpikeSlicer, a novel-design event processing method that can adaptively slice the event streams. To achieve this, SpikeSlicer utilizes an SNN as an event trigger to dynamically determine the optimal moment to split the event stream. Our objectives include: (1) training the SNN to spike at a specific time step for accurate event slicing, and (2) developing a training strategy to identify the best slicing time for a continuous event stream during training. In our paper, we achieve (1) through our newly introduced Spiking Position-aware Loss (SPA-Loss) function, which effectively guides the SNN to spike at the desired time by manipulating the membrane potential. For (2), we implement a Feedback-Update training strategy, where the SNN receives real-time performance feedback from the downstream ANN model for supervision. An overview of our proposed method is depicted in Fig. 2. We evaluate the effectiveness of our proposed SpikeSlicer in two downstream tasks: (i) event-based object tracking, which is strongly sensitive to temporal information and motion dynamics, and (ii) event-based recognition, which is highly related to event density. Extensive experiments validate the effectiveness of the proposed approach.

To sum up, our contributions are as follows:

- We propose SpikeSlicer, a novel plug-and-play event processing method capable of splitting event streams in an adaptive manner.

- We design the SPA-Loss to guide the SNN to trigger spikes at the expected time steps. We then propose a novel Feedback-Update strategy that optimizes the event slicing process based on the ANN feedback.

- Extensive experiments demonstrate that SpikeSlicer significantly improves model performance by up to 11.9% in object tracking and 19.2% in recognition with only a 0.32% increase in energy consumption.

# 2 Background and Related Work

**Event-based Cameras.** They are bio-inspired sensors, which capture the relative intensity changes asynchronously. In contrast to standard cameras that output 2D images, event cameras output sparse event streams. When brightness change exceeds a threshold $C$, an event $e_k$ is generated containing position $\mathbf{u} = (x, y)$, time $t_k$, and polarity $p_k$: $\Delta L(\mathbf{u}, t_k) = L(\mathbf{u}, t_k) - L(\mathbf{u}, t_k - \Delta t_k) = p_k C$. The polarity of an event reflects the direction of the changes (*i.e.*, brightness increase ("ON") or

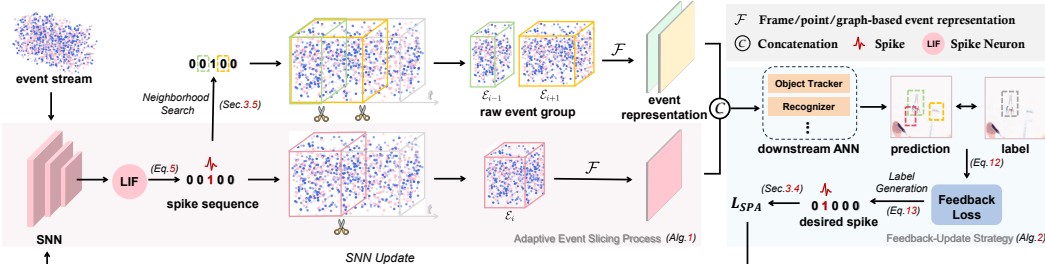

Figure 2: Overview of our method. The input events are first fed into an SNN, and the event is determined to be sliced when a spike occurs. To find the accurate slicing time, the neighborhood search method explores other time steps. and feeds event representations to the downstream ANN model (*e.g.*, object tracker or recognizer). The ANN model then offers feedback, which guides the SNN in firing spikes at the optimal slicing time by supervising the membrane potential through the Spiking Position-aware Loss $\mathcal{L}_{SPA}$.

decrease ("OFF")). In general, the output of an event camera is a sequence of events, which can be described as: $\mathcal{E} = \{e_k\}_{k=1}^N = \{[\mathbf{u}_k, t_k, p_k]\}_{k=1}^N$. With the advantages of high temporal resolution, high dynamic range, and low energy consumption, event cameras are gradually attracting attention in the fields of tracking [3, 15], identification [7], 3D reconstruction [16, 17] and estimation [5].

**Spiking Neural Network (SNN).** SNNs are potential competitors to artificial neural networks (ANNs) due to their distinguished properties: high biological plausibility, event-driven nature, and low power consumption. In SNNs, all information is represented by binary time series data rather than float representation, leading to energy efficiency gains. Also, SNNs possess powerful abilities to extract spatial-temporal features for various tasks, including recognition [18], tracking [2], and image generation [19]. In this paper, we adopt the widely used Leaky Integrate-and-Fire (LIF, [20, 21]) model, which effectively characterizes the dynamic process of spike generation and can be defined as:

$$V[n] = \beta V[n-1] + \gamma I[n], \tag{1}$$

$$S[n] = \Theta(V[n] - \vartheta_{\text{th}}), \tag{2}$$

where $n$ is the time step and $\beta$ is the leaky factor that controls the information reserved from the previous time step; $V[n]$ is the membrane potential; $S[n]$ denotes the output spike which equals 1 when there is a spike and 0 otherwise; $\Theta(x)$ is the Heaviside function. When the membrane potential exceeds the threshold $\vartheta_{\text{th}}$, the neuron will trigger a spike and resets its membrane potential to $V_{\text{reset}} < \vartheta_{\text{th}}$. Meanwhile, when $\beta = \gamma = 1$, LIF neuron evolves into Integrate-and-Fire (IF) neuron. We also introduce a no-reset membrane potential $U[n]$, meaning that the membrane potential does not reset, but directly passes the original value to the next time step (*i.e.*, $U[n] = V[n]$ after Eq. 2).

## 3  Our Approach: SpikeSlicer

In this section, we first introduce the concept of event cells for data preparation (Sec. 3.1). We then introduce the adaptive event slicing process by utilizing an SNN as the event trigger (Sec. 3.2). In Sec. 3.3, we introduce a novel Spiking Position-aware Loss (SPA-Loss) to supervise the SNN to slice the event at the precise time. Lastly, we build a feedback-update (Sec. 3.4) strategy that allows the resulting events to be correlated with the feedback from the downstream model, thereby improving overall performance.

### 3.1  Converting Event Stream to Event Cell

Event streams are asynchronous data that can be represented as a set: $\mathcal{E} = \{[x_i, y_i, t_i, p_i]\}_{i=1}^N$ with a time span of $T$ (*i.e.*, $t_i \in [t_0, t_0 + T]$). We envision an ideal situation where the SNN is utilized to directly process the raw data stream and slice the event. However, in software simulations, the event stream should be converted into event representations to comply with the input requirements of existing deep learning frameworks. In this paper, following the mainstream research, we adopt the voxel-grid representation [22] as the input of SNN. We first introduce the definition of event cell:

**Definition 1** (Event cell). *Consider a small time interval $\delta t$, event cell is a single-grid event representation in the form of: $C_\pm(x, y, t_*) = \mathcal{F}_{\text{voxel}}(G_\pm(x, y, t, \{t \in [t_*, t_* + \delta t]\}))$, where $\mathcal{F}_{\text{voxel}}$ denotes the voxel grid [22] method to process the raw event groups $G_\pm$ (Appendix F) with $t \in [t_*, t_* + \delta t]$ into a grid representation.*

A whole event stream can be then represented by a list of $N$ event cells, *i.e.*, $\{C_\pm(x, y, t_0), C_\pm(x, y, t_0 + \delta t), ..., C_\pm(x, y, t_0 + (N-1)\delta t)\}$, where $N = T/\delta t$ and each cell corresponds to a discrete time index $n \in \{0, 1, \cdots, N-1\}$. The mapping of discrete time to real event time interval is defined as:

$$f_{\text{time}}(n) = \{t | t \in [t_0 + n\delta t, t_0 + (n+1)\delta t]\}. \tag{3}$$

*e.g.*, the time interval of $C_\pm(x, y, t_0)$ is $f_{\text{time}}(0)$. In the following sections, we abbreviate the event cell as $C[n]$.

**Discussion:** Distinct from the typical voxel grid, an event cell only contains a brief time interval, *i.e.*, $\delta t$ is small. At this point, the entire cell sequence appropriately represents the raw event stream, while simultaneously fulfilling the input requisites for the SNN.

## 3.2 Adaptive Event Slicing Process

Utilizing SNNs on neuromorphic hardware for processing event streams is low-energy and low-latency [23, 24]. Hence, we adopt the Spiking Neural Network as the event stream slicer, aiming for dynamically slicing the event stream to enhance the downstream performance. Incorporating with the SNN, we now describe the adaptive event slicing process:

Considering an event stream $\mathcal{E}$, we first convert $\mathcal{E}$ into a list of time-continuous event cells. Event cells are then continuously entered into a SNN (structure details are provided in Appendix L) through a loop operation. Through forward propagation, the features of the last hidden layers ($h^{L-1}$) are finally mapped to a single spiking neuron to activate spikes:

$$S_{\text{out}} = \text{LIF}(\text{SNN}_{\text{FC}}(h^{L-1})). \tag{4}$$

Once the spiking neuron generates a spike (*i.e.*, $S_{out} = 1$) at current time $n_c$, it is considered a slicing action. This allows us to obtain the time interval from the end of the previous spike to the current spike. Suppose the previous spike happened at time $n_p$, the real event time interval is within $T_{\text{group}} = \bigcup_{n=n_p+1}^{n_c} f_{\text{time}}(n) = \{t \in [t_0 + (n_p+1)\delta t, t_0 + (n_c+1)\delta t]\}$. Then, the raw event data within this time interval form an event group, which can be converted to any event representation:

$$\mathcal{D}_{n_c} = \mathcal{F}(G_\pm(x, y, t, T_{\text{group}})), \tag{5}$$

where $\mathcal{F}$ denotes an event representation method (*e.g.*, frame [22], point [25], graph [26] and surface [6]-based methods). This representation $\mathcal{D}_{n_c}$ is then prepared for the downstream tasks.

**Event Slicing Rule:** The slice of the event stream is determined by the state (excited/resting) of the SNN's spiking neuron. Serving as a dynamic event trigger, SNN promptly decides to split events upon spike generation and extracts the precise time interval of the raw event stream. The details of the adaptive event slicing process is shown in Alg. 1.

## 3.3 Spiking Position-aware Loss

In this section, we propose the Spiking Position-aware Loss (SPA-Loss), which contains two parts: (1) membrane potential-driven loss (Mem-Loss) is used to directly guide the spiking state of the spiking neuron at a specified timestamp, and (2) linear-assuming loss (LA-Loss), which is designed to resolve the dependence phenomenon between neighboring membrane potentials, achieving a more precise spiking time. Moreover, we introduce a (3) dynamic hyperparameter tuning method to avoid the experimental bias caused by the manual setting of hyperparameters.

### 3.3.1 Membrane Potential-driven Loss

As mentioned in the previous section, the slice position of event is determined upon the spike occurrence. The challenge now lies in directing the SNN to trigger a spike precisely at the optimal position, once the label for this optimal slicing position is provided (in Sec. 3.4).

Consider consecutive event cells as inputs starting from the previous spiking time, suppose we expect SNN to slice the event at $n^*$, *i.e.*, a spike $S_{out}$ is triggered at $n^*$. This corresponds to the membrane potential of the spiking neuron needing to reach the threshold $V_{th}$ at $n^*$, which inspired us to guide the spike time by directly giving the desired membrane potentials. However, membrane potential returns to the resting state immediately after the occurrence of a spike, which may result in inaccurate guidance at later moments (Appendix G). Thus, we choose to supervise the no-reset membrane

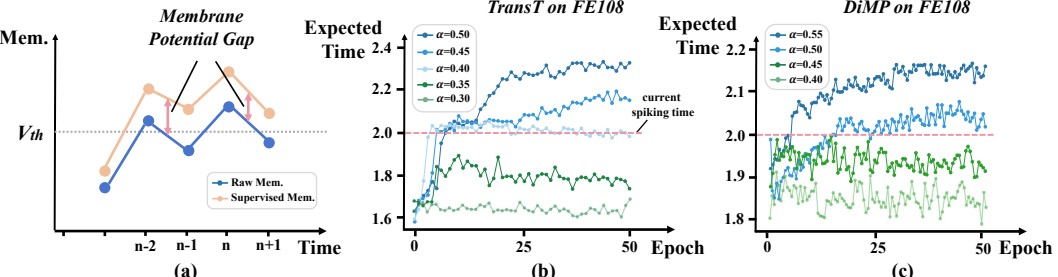

Figure 3: Empirical observations: (a) Hill effect in adaptive slicing process; (b) Impact of hyperparameter $\alpha$ settings on TransT tracker [27] and (c) DiMP tracker [28].

potential $U[n]$ (Eq. 16) to exceed the threshold (*i.e.*, $U[n^*] \geq V_{th}$). The membrane potential-driven loss is defined as:

$$\mathcal{L}_{Mem} = \left|\left|U[n^*] - (1+\alpha)V_{th}\right|\right|_2^2, \tag{6}$$

where $\alpha \geq 0$ is a hyperparameter to control the desired membrane potential to exceed the threshold. However, an excessively high $\alpha$ may directly induce a premature spike in the neuron, thereby influencing the membrane potential state at the targeted time step. We provide a proposition to address this problem:

**Proposition 1.** *Suppose the input event cell sequence has length $N$, desired spiking time is $n^*$ ($n^* \in \{0, 1, ..., N-1\}$), the membrane potential at time $n^*$ satisfying the constraints:*

$$V_{th} \leq U[n^*] \leq \max(\beta V_{th} + \gamma I[n^*], V_{th}), \tag{7}$$

*where $I[n^*]$ is the input synaptic current from Eq.1. Then the spiking neuron fires a spike at time $n^*$ and does not excite spikes at neighboring moments.*

The proof is provided in the Appendix H. Based on the proposition, we modify the loss function into:

$$\mathcal{L}_{Mem} = \left|\left|U[n^*] - \left((1-\alpha)U_{\text{lower}} + \alpha U_{\text{upper}}\right)\right|\right|_2^2, \tag{8}$$

where $U_{\text{lower}} = V_{th}$ and $U_{\text{upper}} = \max(\beta V_{th} + \gamma I[n^*], V_{th})$ denote the lower and upper bounds of the $U[n^*]$ provided in the proposition, respectively; $\alpha \in [0, 1]$ balances the desired membrane potential $U[n^*]$ between $U_{\text{lower}}$ and $U_{\text{upper}}$. Experiments in Sec. 4.1 demonstrate that Mem-Loss is able to supervise the SNN to determine the slicing of the event flow at a specified timestamp. More details of Mem-Loss are provided in Appendix I.1.

### 3.3.2 Linear-assuming Loss

However, only using Mem-Loss is unable to guarantee that the spiking neuron can trigger spikes at any expected timestamp. We have the following observations:

**Observation 1** (Hill effect). Suppose there exists a situation where $S[n] = 1$ and $U[n] \geq U[n+1]$. If the neuron is expected to activate a spike at time $n+1$, $U[n+1]$ will be driven to reach the threshold through the Mem-Loss. Nonetheless, the supervised neuron still exhibits $U[n] \geq U[n+1]$, causing an early spike at time $n$.

As illustrated in Fig. 3(a), if $U[n] \geq U[n+1]$ exists, this membrane potential gap is still inherited after the supervision. In this case, when the later membrane potential is guided to exceed the threshold, the earlier membrane potential reaches the threshold sooner and turns the neuron into the resting state. This poses challenges in obtaining a second spike at the later moment.

Therefore, we expect the later membrane potential to increase monotonically with the time step to reverse the hill effect. Here we use the simplest linear monotonically increasing assumption to construct the loss function:

$$\mathcal{L}_{LA} = \begin{cases} ||U[n_c] - V_{th}\, \dfrac{n_c}{n^*}||_2^2, & \text{if condition}_n; \\ 0, & \text{otherwise,} \end{cases} \tag{9}$$

where $n^*$ denotes the expected spike timestep and $n_c$ denotes the current spike timestep, $\text{condition}_n$ corresponds to the nonlinear monotonically increasing condition that satisfies: $U[n_c] \geq$

| **Algorithm 1** Adaptive Event Slicing Process | **Algorithm 2** Feedback-Update Training Strategy |
|---|---|
| **Input:** SNN model, input event $\mathcal{E}$, the number of event cell $N$, the previous spike index $n_p$, list $\mathcal{D}_{\text{rep}}$.
**Initialize:** $n_p = 0$.
**for** all $n = 0, 1, 2, ... N - 1$ **do**
$\quad S_{\text{out}} = \text{SNN}(C[n])$.
$\quad$**if** $S_{\text{out}} == 1$ **then**
$\quad\quad n_c = n$.
$\quad\quad T_{\text{group}} = \bigcup_{n=n_p+1}^{n_c} f_{\text{time}}(n)$.
$\quad\quad \mathcal{E}_{\text{group}} = G_{\pm}(x, y, t, T_{\text{group}})$.
$\quad\quad \mathcal{D}_{n_c} = \mathcal{F}(\mathcal{E}_{\text{group}})$.
$\quad\quad$Append $\mathcal{D}_{n_c}$ to $\mathcal{D}_{\text{rep}}$.
$\quad\quad n_p = n_c + 1$.
$\quad$**end if**
**end for**
**Return:** $\mathcal{D}_{\text{rep}}$. | **Input:** SNN model, pretrained ANN model, ANN training dataset $\mathcal{D}_A$, SNN training dataset $\mathcal{D}_S$, total epoch $E_{train}$, epoch to start finetuning $e_f$, alpha $\alpha$, learning rate $\eta$.
**for** all $e = 1, 2, ... E_{train}$ epoch **do**
$\quad$**for** all event batch $d_i = d_1, d_2, ... d_{N_s}$ in $\mathcal{D}_S$ **do**
$\quad\quad$Feed $d_i$ into SNN until it spikes at time step $n_c$.
$\quad\quad$Generate event candidates to ANN to get feedback.
$\quad\quad$Calculate loss function $\mathcal{L}_{\text{SPA}} = \mathcal{L}_{\text{Mem}} + \mathcal{L}_{\text{LA}}$.
$\quad\quad$Backpropagate and update SNN parameters.
$\quad$**end for**
$\quad \alpha \leftarrow \alpha - 2 \cdot \eta \sum_i^{N_s} (n^{*i} - n_c^i)/N_s$.
$\quad$**if** $e > e_f$ **then**
$\quad\quad$Split $\mathcal{D}_A$ into $\mathcal{D}'_A$ by adaptive event slicing process.
$\quad\quad$Finetune ANN parameters on $\mathcal{D}'_A$.
$\quad$**end if**
**end for** |

$U[n^*]$ & $n_c < n^*$. We expect the membrane potential at $n_s$ to reach $\frac{n_c}{n^*} V_{th}$ for the latter membrane potential at the $n^*$ to reach $V_{th}$ in a linearly increasing form. More explanations are provided in Appendix I.2.

Combined with Mem-Loss and LA-Loss, we defined the SPA-Loss, which guides the adaptive event slicing process in subsequent experiments: $\mathcal{L}_{SPA} = \mathcal{L}_{Mem} + \mathcal{L}_{LA}$.

### 3.3.3 Dynamic Hyperparameter Tuning

Although controlling the SNN to spike at a desired location can be achieved through the combination of Mem-Loss and LA-Loss, the utilization of varying $\alpha$ values (Eq. 8) may result in significant fluctuations in experimental results. We have the following observation.

**Observation 2.** The larger the $\alpha$, the earlier the SNN tends to fire spikes; and vice versa.

A larger $\alpha$ in Eq. 8 implies a higher pre-momentary membrane potential, which results in an earlier spike. Taking the larger-$\alpha$ scenario in Fig. 3(c) as an example, if the SNN is expected to activate a spike at a later timestep, the larger $\alpha$ prevents the actual spike from being delayed. Thus, we need to decrease $\alpha$, causing the expected spike time to shift earlier. This concludes that the update direction of the hyperparameter $\alpha$ should be consistent with the update direction of the desired spiking index.

**Observation 3.** A fixed $\alpha$ leads to significant variations in performance across different tasks.

As illustrated in Fig. 3(b) and (c), the same $\alpha$ varies significantly on different downstream models, which makes it difficult to set the hyperparameter alpha in advance.

To address the above issues, we design a learning-based hyperparameter tuning method for updating $\alpha$ (in Alg. 2). More details are provided in Appendix I.3.

### 3.4 Feedback-Update Strategy through SNN-ANN Cooperation

Based on the methods proposed in the previous sections: if the desired trigger time $n^*$ is given, the SNN is able to accurately accomplish the event slicing under the guidance of the SPA-Loss function. In this section, we focus on how to obtain the desired spike time $n^*$ through the downstream ANN feedback. We thus propose a feedback-update strategy that enables the SNN to slice events when the downstream ANN model achieves optimal performance. By receiving real-time feedback from the downstream model and updating $n^*$, this strategy ultimately enhances task performance.

Particularly, when SNN processes the input event and triggers a spike at time $n_c$, it returns a spike output sequence $S = [0, ...0, 1, 0, ...]$, where 1 is at $n_c$-th. We first perform a *neighborhood search* to obtain $2d + 1$ candidate event representation with the index in $\{n_c - d, ..., n_c + d\}$: $\{D_{n_c-d}, ..., D_{n_c+d}\}$, where $D_{n_c+i} = \mathcal{F}(G_{\pm}(x, y, t, \{t \in [t_0 + (n_p + 1)\delta t, t_0 + (n_c + 1 + i)\delta t]\}))$. We then choose a downstream model $\mathcal{M}$ (*e.g.*, object tracker or recognizer) and feed the candidate event representations into it to obtain feedback $y$:

$$y = \mathcal{L}_{\mathcal{M}}(C[n_c - d]) \oplus ... \oplus \mathcal{L}_{\mathcal{M}}(C[n_c + d]), \qquad (10)$$

where $\mathcal{L}_{\mathcal{M}}(\cdot)$ returns the output loss of $\mathcal{M}$ and $\oplus$ concatenates these losses into $y \in \mathbb{R}^{2d+1}$. We choose the model loss as the feedback since it directly reflects the quality of inputs. We can then generate the desired spike index $n^*$ by: $n^* = \arg\min_i(y[i])$, where $\arg\min$ extracts the index with

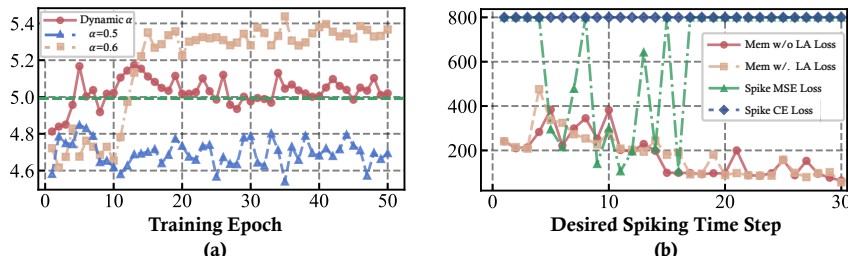

Figure 4: (a) Experiments on comparing different loss functions on a simple event slicing task. Our proposed Mem-Loss and LA-Loss require only a small number of iterations to supervise the SNN to activate spikes at the desired time steps; (b) Experiments on different hyperparameter settings. Our dynamic tuning method can stably converge towards the optimal spiking time (colored in green). In contrast, using a fixed $\alpha$ results in unstable training and challenges in finding the optimal point.

the best (minimal loss) feedback, which in turn guides the dynamic slicing process using SPA-Loss. After training the SNN, the ANN is then updated by feeding the newly-sliced events, thus forming an SNN-ANN cooperation process.

**Feedback-Update Strategy.** This strategy employs a two-stage iterative approach. In the first stage, the ANN offers real-time feedback to the adaptive slicing process for training SNN. In the second stage, the trained SNN provides a newly-sliced event to finetune the ANN. The process then iterates back to the first stage. This strategy provides a novel SNN-ANN cooperation paradigm which establishes a strong connection between raw data and the downstream model. We summarize the feedback-update strategy in Alg. 2.

## 4 Experiments

To evaluate the effectiveness of our proposed method, we set up two-level experiments. In the beginner's arena, we expect the SNN to find the exact slicing time with the simulated event inputs. In the expert's arena, we conduct experiments on event-based object tracking and image recognition. Details of experiment settings and more experimental analyses are presented in the Appendix.

### 4.1 Beginner's Arena: Event Slicing in Simple Tasks

We first conduct some entry-level tasks to validate the effectiveness of SPA-Loss. We set up the task: *Input $N$ randomized event cells, expect the SNN to slice at a specified time step $n^*$ and there exists a certain probability of interfering with SNN to slice at other time steps.*

We compare the SPA-Loss function with common MSE-Loss and CE-Loss. The experiment setting is detailed in Appendix K. As depicted in Fig 4(b), the SPA-Loss successfully supervises the SNN to activate spikes at the desired time steps. In particular, SPA-Loss requires only a few iterations ($<400$) to supervise the SNN to fire spikes correctly. In contrast, MSE-Loss can only succeed at certain time steps, and CE-Loss cannot even accomplish the task. In addition, using both Mem-Loss and LA-Loss yields smoother results compared to using Mem-Loss alone. To summarize, the beginner's arena preliminarily tests the effectiveness of the SPA-Loss and paves the way for subsequent experiments.

### 4.2 Expert's Arena: Mastering Adaptive Event Slicing with SNN-ANN Collaboration

After a successful challenge in the beginner's arena, we move on to the expert arena. Here we use a low-energy SNN to adaptively process the event data on complex downstream tasks:

**Event-based Object Tracking.** Since the tracking task is highly sensitive to temporal information, dynamic event slicing is of great importance. We provide two versions for adaptive slicing: SpikeSlicer-Base (B) and SpikeSlicer-Small (S). The detailed consumption of these two versions are introduced in the ablation study. Tab. 1 shows that the tracking performances under the SpikeSlicer have a significant improvement in terms of representative success rate (RSR), representative precision rate (RPR), and overlap precision (OP). For instance, TransT with SpikeSlicer-S's performance on $OP_{0.50}$ and $OP_{0.75}$ improved by 5.8% and 3.2% compared to its original result under the HDR scenario. When compared to results on the fixed event (the number of the fixed-sliced event is aligned with the number of dynamic-sliced events to ensure a fair comparison), our method achieves favorable gains in the overall RSR, *i.e.*, 63.6 *vs.* 51.0.

Table 1: Quantitative comparison on FE108. There are four challenging scenarios, including high dynamic range (HDR), low light (LL), fast motion with and without motion blur (FWB & FNB) and all testing datasets (ALL). [*method*]+*SpikeSlicer-B/S* represents the results based on our adaptive event slicing method with base (B) or small (S) SNN version. The results of [*method*] are reproduced on the original fixed-sliced event dataset. To ensure a fair comparison, *fix event* indicates that the model is tested on a dataset of fixed-sliced event frames, where the number of fixed event frames is the same as the number of dynamically sliced event frames by using SpikeSlicer. Best performances are denoted by deep green.

| Methods | HDR | | | | LL | | | | FWB | | | | FNB | | | | ALL | | | |
|---|---|---|---|---|---|---|---|---|---|---|---|---|---|---|---|---|---|---|---|---|
| | RSR | OP$_{.50}$ | OP$_{.75}$ | RPR | RSR | OP$_{.50}$ | OP$_{.75}$ | RPR | RSR | OP$_{.50}$ | OP$_{.75}$ | RPR | RSR | OP$_{.50}$ | OP$_{.75}$ | RPR | RSR | OP$_{.50}$ | OP$_{.75}$ | RPR |
| SiamFC++ [29] | 15.3 | 15.0 | 1.3 | 25.2 | 13.4 | 8.7 | 0.8 | 15.3 | 28.6 | 36.3 | 6.0 | 48.2 | 36.8 | 42.7 | 7.4 | 63.1 | 23.8 | 26.0 | 3.9 | 39.1 |
| KYS [30] | 15.7 | 14.5 | 5.2 | 23.0 | 12.0 | 8.0 | 1.1 | 18.0 | 47.0 | 63.9 | 14.8 | 73.3 | 36.9 | 44.5 | 15.2 | 57.9 | 26.6 | 30.6 | 9.2 | 41.0 |
| CLNet [31] | 30.0 | 33.5 | 9.6 | 48.3 | 13.7 | 6.0 | 0.9 | 23.6 | 52.9 | 71.2 | 23.3 | 80.3 | 40.8 | 46.3 | 14.2 | 67.7 | 34.4 | 39.1 | 11.8 | 55.5 |
| DiMP [28] | 49.1 | 60.3 | 16.3 | 77.1 | 67.3 | 87.4 | 40.4 | 96.9 | 52.5 | 53.9 | 7.8 | 98.5 | 50.0 | 60.1 | 21.4 | 78.2 | 52.1 | 62.4 | 17.9 | 84.3 |
| DiMP (*fixed event*) | 53.3 | 68.2 | 21.4 | 81.6 | 67.6 | 86.3 | 43.1 | 95.0 | 49.7 | 45.4 | 5.88 | 80.5 | 49.6 | 59.4 | 23.7 | 75.3 | 53.8 | 64.3 | 19.5 | 82.4 |
| DiMP+**SpikeSlicer-B** | 53.3 | 67.3 | 21.9 | 79.8 | 69.8 | 92.3 | 47.2 | 96.5 | 64.1 | 83.4 | 27.9 | 97.2 | 54.4 | 67.4 | 27.7 | 81.2 | 57.3 | 73.0 | 25.8 | 86.1 |
| DiMP+**SpikeSlicer-S** | 56.0 | 72.0 | 27.0 | 80.7 | 70.0 | 92.2 | 50.6 | 95.0 | 66.7 | 86.1 | 38.4 | 96.9 | 56.4 | 70.6 | 31.1 | 81.4 | 59.6 | 76.8 | 30.9 | 86.4 |
| PrDiMP [32] | 50.3 | 62.2 | 19.4 | 77.8 | 68.8 | 90.4 | 41.9 | 97.0 | 56.6 | 68.8 | 11.2 | 98.1 | 53.4 | 64.7 | 23.4 | 82.7 | 54.5 | 67.4 | 20.4 | 85.8 |
| PrDiMP (*fixed event*) | 41.2 | 49.8 | 18.4 | 66.1 | 42.7 | 45.2 | 12.5 | 87.1 | 62.4 | 85.8 | 21.3 | 90.4 | 47.6 | 58.3 | 18.5 | 77.0 | 48.0 | 61.2 | 19.2 | 78.6 |
| PrDiMP+**SpikeSlicer-B** | 53.7 | 67.1 | 22.2 | 80.2 | 70.1 | 94.5 | 41.8 | 97.2 | 70.7 | 89.2 | 54.2 | 93.9 | 56.1 | 70.3 | 25.7 | 83.7 | 59.2 | 75.3 | 29.1 | 86.8 |
| PrDiMP+**SpikeSlicer-S** | 55.2 | 70.7 | 24.9 | 79.9 | 71.3 | 95.7 | 45.1 | 97.7 | 72.5 | 91.4 | 59.2 | 95.4 | 57.6 | 73.0 | 27.6 | 83.8 | 60.9 | 78.2 | 32.3 | 87.2 |
| TaMOs [33] | 37.9 | 43.5 | 1.5 | 66.9 | 46.4 | 30.5 | 0.2 | 87.8 | 50.8 | 54.2 | 1.0 | 96.7 | 40.9 | 36.2 | 0.7 | 77.1 | 42.5 | 41.2 | 1.0 | 78.8 |
| TaMOs (*fixed event*) | 44.0 | 57.0 | 3.9 | 72.7 | 49.5 | 48.0 | 0.3 | 90.0 | 46.2 | 34.0 | 0.5 | 69.7 | 43.4 | 46.0 | 1.5 | 78.4 | 45.1 | 48.5 | 2.1 | 77.1 |
| TaMOs+**SpikeSlicer-B** | 40.4 | 45.4 | 1.7 | 72.2 | 47.1 | 34.6 | 0.1 | 91.7 | 48.7 | 45.7 | 0.3 | 98.5 | 43.6 | 42.0 | 1.0 | 81.7 | 44.2 | 43.1 | 1.1 | 83.3 |
| TaMOs+**SpikeSlicer-S** | 41.4 | 48.1 | 1.4 | 71.7 | 48.5 | 36.1 | 0.3 | 95.5 | 45.0 | 18.7 | 0.1 | 98.7 | 40.7 | 29.6 | 0.5 | 80.4 | 43.0 | 36.3 | 0.8 | 82.9 |
| TransT [27] | 55.8 | 69.4 | 27.7 | 82.0 | 70.5 | 90.9 | 49.7 | 98.3 | 74.1 | 98.5 | 55.5 | **99.9** | 58.6 | 73.9 | 30.2 | 87.2 | 61.3 | 78.3 | 33.8 | 89.2 |
| TransT (*fixed event*) | 51.4 | 67.8 | 11.1 | 81.2 | 63.2 | 80.2 | 28.3 | 89.3 | 41.5 | 28.0 | 2.50 | 57.7 | 50.6 | 57.9 | 12.7 | 78.9 | 51.0 | 59.0 | 12.0 | 78.8 |
| TransT+**SpikeSlicer-B** | 58.0 | 73.4 | 29.4 | 83.8 | 71.9 | 95.9 | 47.7 | **99.3** | 73.4 | 95.7 | 56.9 | 96.2 | **60.5** | **76.6** | 32.0 | **88.4** | 62.1 | 79.9 | 34.6 | 88.7 |
| TransT+**SpikeSlicer-S** | **59.1** | **75.2** | **30.9** | **84.4** | **72.9** | **97.9** | **52.2** | 99.1 | **76.6** | **99.5** | **65.7** | 99.8 | 60.2 | 76.5 | **32.3** | 87.5 | **63.6** | **82.3** | **37.5** | **90.1** |

Table 2: Quantative comparison on DVS-Gesture, N-Caltech101, DVS-CIFAR10 and SL-Animals. *Random* and *Fix* denote that the input events are randomly sliced and fixed sliced, respectively. Instead, our method slices the event stream adaptively.

| Method | DVS-Gesture [34] | | | N-Caltech101 [35] | | | DVS-CIFAR10 [36] | | | SL-Animals [37] | | |
|---|---|---|---|---|---|---|---|---|---|---|---|---|
| Slice Type | Random | Fix | Ours | Random | Fix | Ours | Random | Fix | Ours | Random | Fix | Ours |
| ResNet-18 | 93.06 | 93.40 | **94.79** | 77.80 | 73.37 | **79.86** | 78.91 | 77.73 | **81.15** | 85.71 | 83.93 | **88.39** |
| ResNet-34 | 95.14 | 93.40 | **96.18** | 78.77 | 76.08 | **82.54** | 80.57 | 79.39 | **82.23** | 86.61 | 87.50 | **89.93** |
| Swin-S | 88.19 | 89.93 | **91.67** | 86.30 | 81.76 | **87.30** | 81.54 | 79.79 | **83.01** | 74.11 | 56.25 | **75.45** |

**Event-based Recognition.** We also conduct experiments in event-based recognition to evaluate the effectiveness of our proposed method. As depicted in Tab. 2, our method has a significant improvement over the fixed-sliced method, with an accuracy improvement of 2.78% and 6.46% by using ResNet-34 in DVS-Gesture and N-Caltech101, respectively. To verify that the results of our adaptive slicing method are not biased due to randomness, we add the random-slice baselines for comparison, in which the event stream is randomly sliced into event groups and fed into the ANN for training. Our method also yields better performance compared with the random-slice results.

**Visualization of Adaptive Event Slicing.** We visualize the tracking results to demonstrate that the dynamic slicing method is able to adapt to various motion scenarios. As shown in Fig. 5, our method obtains better tracking performance compared to fixed event inputs, *i.e.*, the position of the prediction box is more accurate. Additionally, our dynamic event slicing method can achieve (1) *edge enhancement* and (2) *redundancy removal* to refine the event data under different tracking scenarios. However, the fixed-slice approach adopts the same slicing strategy for each event stream, leading to performance degradation.

### 4.3 Analysis of the Adaptive Slicing Method

**Analysis of Spike Splitting Time and Event Density.** To evaluate the effectiveness of our proposed method, we conducted a detailed visualization analysis, as depicted in Fig. 6, examining the relationship between the locations of split points and the corresponding event stream densities. The definition of event density is detailed in the Appendix B. The analysis reveals a clear **match** between the positions of cuts made by SNN and the respective event density. Specifically, the SNN tends to perform more frequent cuts in regions of higher event density, while conversely, regions with lower event density experienced fewer cuts. These findings indicate that the dynamic cutting method is effectively adaptive to the varying information density within the event stream.

Table 3: Comparison of Efficiency and Speed. The comparison includes the number of operations (OPs) and tracking speed per image without image processing time.

| Models | OPs (G) | Energy (mJ) | Speed (s / img) | Performance |
|---|---|---|---|---|
| **ANN w/o SpikeSlicer** | 56.36 | 259.26 | 0.045 | 51.0 |
| **ANN w/ SpikeSlicer** | 57.09 | 260.11 | 0.060 | 62.4 |

Table 4: Experiments on different event representations with fixed or dynamic slicing methods. Our method yields significant improvement when using different event representation methods, proving SpikeSlicer's effectiveness as a plug-and-play event slicer.

| FE108 | Time Surface | | Event Spike Tensor | | Voxel Grid | |
|---|---|---|---|---|---|---|
| *Slice Method* | *RSR* | *RPR* | *RSR* | *RPR* | *RSR* | *RPR* |
| Fix Slice | 57.5 | 85.7 | 50.4 | 81.6 | 51.0 | 78.8 |
| **SpikeSlicer (ours)** | **59.5** | **86.8** | **54.5** | **85.6** | **62.4** | **88.9** |
| **DVS-Gesture** | **Event Frame** | | **Event Spike Tensor** | | **Voxel Grid** | |
| *Slice Method* | *Accuracy* | | *Accuracy* | | *Accuracy* | |
| Fix Slice | 93.75% | | 93.75% | | 88.54% | |
| **SpikeSlicer (ours)** | **94.79%** | | **95.49%** | | **89.24%** | |

Table 5: Ablation studies for evaluating the proposed loss function on the SL-Animals dataset.

| Slice Method | $\mathcal{L}_{Mem}$ | $\mathcal{L}_{LA}$ | ResNet-18 | ResNet-34 |
|---|---|---|---|---|
| Fix Slice | ✗ | ✗ | 83.93 | 87.50 |
| SpikeSlicer | ✓ | ✗ | 87.50 (+3.57) | 88.52 (+1.02) |
| SpikeSlicer | ✓ | ✓ | 88.39 (+4.46) | 89.73 (+2.23) |

Table 6: Experiments with different event cell numbers $N$. The resulting sliced event group always has a similar time interval in various $N$ conditions.

| $N$ | 15 | 20 | 25 |
|---|---|---|---|
| Avg Spike Time | 2.42 | 3.15 | 4.77 |
| Time Duration | 16.13% | 15.75% | 19.08% |

**Analysis of Efficiency and Speed.** We evaluate the performance of the ANN tracker (TransT) with or without the SpikeSlicer in terms of energy consumption and processing speed. Detailed calculations for energy consumption are in Appendix M. In Tab. 3, the SpikeSlicer stands out as a plug-and-play option. The introduction of the SpikeSlicer incurs a marginal energy increase of 0.85 mJ and a small reduction in processing speed. However, these costs are offset by a significant 22.3% improvement in the RSR metric.

**Evaluation of the Dynamic Hyperparameter Tuning.** We use different $\alpha$ settings in tracking experiments to study the effect of hyperparameters. Fig. 4(b) shows that the fixed $\alpha$ setting does not allow the spiking time step to reach the desired time step accurately, while our dynamic $\alpha$ tuning allows SNN to spike at the desired time step, leading to a better event-slicing process.

### 4.4 Ablation Study

**Ablation Study on Event Representation Method.** Since SpikeSlicer works as a plug-and-play event slicing method suitable for multiple event representations, we thus conduct our method through different event representation methods to further evaluate the effectiveness. Tab. 4 demonstrates that, across various forms of event representation, the dynamic splitting method yields performance enhancements in both tracking and recognition tasks when compared to the fixed splitting method.

**Ablation Study on Different Components of SPA-Loss.** The results in Tab. 5 reveal that the adaptive slicing method achieves accuracy improvement, and all of our proposed loss functions (including $\mathcal{L}_{Mem}$ and $\mathcal{L}_{LA}$) contribute to the performance, showing the effectiveness of our method.

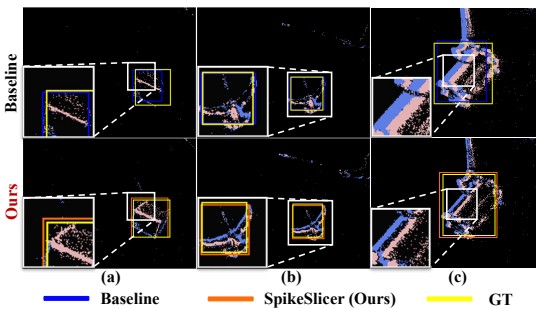

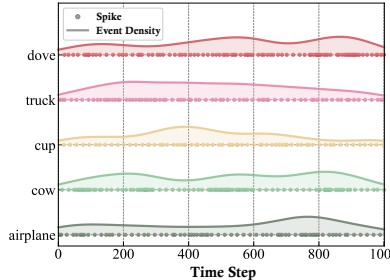

Figure 5: Visualization results on FE108 dataset. The white box denotes the zoom-in area. Our adaptive event slicing method provides better tracking performance than fixed counterparts while enabling edge enhancement (a,b) and redundancy removal (c).

Figure 6: Visualization of spike splitting points with corresponding event density. The results of our dynamic splitting are **matched** to real event information. The higher the event density, the SNN splits more frequently; conversely, the SNN splits more sparsely.

Table 7: Ablation studies for different network sizes of the SpikeSlicer. The comparison includes the model consumptions and tracking performances of PrDiMP.

| Methods | Model Consumptions | | | Performances | | | |
|---|---|---|---|---|---|---|---|
| | *Params (M)* | *OPs (G)* | *Energy (mJ)* | *RSR* | *OP$_{.50}$* | *OP$_{.75}$* | *RPR* |
| **SpikeSlicer-Base** | 45.11 | 0.73 | 0.85 | 59.24 | 75.25 | 29.12 | 86.82 |
| **SpikeSlicer-Small** | 0.42 | 0.56 | 0.69 | 60.88 | 78.19 | 32.34 | 87.19 |

**Ablation Study on Event Cell Number.** Considering that the number of event cells $N$ may affect the SNN's decision on event slicing, we examine the stability of the sliced event group by varying the size of $N$. Tab. 6 shows that the average spike time of the SNN varies for different $N$, but the time duration (*i.e.*, Avg Spike/$N$) of the resulting event groups is stable. This verifies that the SNN can effectively make cuts based on event information rather than making decisions based on the number of inputs alone. More details are provided in Appendix P.

**Ablation Study on Different Network Sizes of the SpikeSlicer.** To evaluate the performance of SpikeSlicer under different network sizes, we conduct an additional ablation study (in Tab. 7) with a smaller variant, SpikeSlicer-Small. This lightweight model contains only 0.42M parameters—significantly fewer than the base model's while achieving comparable or better performance across key metrics. This compact design demonstrates its potential for efficient deployment on hardware platforms, providing a strong foundation for real-world applications requiring lightweight neural networks.

## 5 Limitation

We summarize the limitations of this work as follows: Firstly, the SpikeSlicer process involves multi-stage SNN-ANN training, which leaves substantial room for improvement in the adaptive slicing strategy. Secondly, for recognition tasks, we convert the event stream into a single-frame representation to obtain accurate supervisory signals. This approach could be refined in the future to enable SpikeSlicer to slice stream events into multi-frame representations, which are the mainstream format. Thirdly, our experiments are conducted on GPUs; however, the most suitable hardware for SNNs would be brain-inspired chips. In addition, dynamic events need to be generated and processed in real-time during inference, rather than fixed generation in advance. As a result, conducting experiments on GPUs may lead to slower overall inference speeds. Extending this paradigm to brain-inspired chips with asynchronous event input is an interesting direction worth exploring in the future.

## 6 Conclusion

In this work, we proposed SpikeSlicer, a novel event processing method that splits event streams adaptively. SpikeSlicer utilizes a spiking neural network (SNN) as an event trigger, which determines the slicing time according to the generated spikes. To achieve accurate slicing, we designed the Spiking Position-aware Loss (SPA-Loss) which guides the SNN to trigger spikes at the desired time step. In addition, we proposed a Feedback-Update training strategy that allows the SNN to make accurate slicing decisions based on the ANN feedback. Extensive experiments have demonstrated the effectiveness of SpikeSlicer in yielding performance improvement in event-based object tracking and recognition tasks. In the future, we will assess SpikeSlicer's suitability for other event-based tasks, and devise more efficient training strategies for the SNN-ANN cooperative framework to optimize real-time processing in the future.

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

# Appendix

## A Details of our Motivations

To clarify the motivation behind our dynamic event stream slicing algorithm, this section tells the details.

### A.1 Motivation for Proposing a Dynamic Event Stream Slicing Algorithm

The process of event-to-representation conversion is mainly divided into two steps: **Step 1. Slice the raw event stream into multiple sub-event stream**, and **Step 2. convert these sub-event streams into representations using various event representation methods.** While much work has focused on optimizing event representation (Step 2) to extract better event information, including time surface and EST, they do not address the issues arised with fixed slicing (*e.g.*, resulting non-uniform event in scenarios with changing motion speed). Despite event slicing being a small part of the overall pipeline, it is a critical point. This is because the event stream is very sensitive to slicing, and the model performance fluctuates very much for different slicing methods, as proved by extensive experiments in Appendix C.

To better address this issue, we introduced the dynamic slicing method SpikeSlicer. Meanwhile, SpikeSlicer is guided by downstream task feedback to ensure that the new sub-streams could enhance downstream task performance.

### A.2 Motivation for for Using SNN as a Slicing Trigger

The reason why we choose SNN as the event slicing trigger is twofold:

- Utilizing SNNs on neuromorphic hardware for processing event streams is low-energy and low-latency [23, 24].

- Deployed on neuromorphic hardware, SNNs can process event streams asynchronously [38–40], conserving energy when there is no data input—a capability that GPUs, operating synchronously, lack.

Due to the aforementioned reasons, there is a considerable amount of research [41, 42, 11] employing Spiking Neural Networks (SNNs) for event data. Although these SNNs are simulated on GPU platforms, the models resulting from such simulations could be deployed on neuromorphic hardware.

### A.3 Contribution for Using SNN as a Slicing Trigger

We propose a new cooperative paradigm where SNN acts as an efficient, low-energy data processor to assist the ANN in improving downstream performance. This is a brand-new SNN-ANN cooperation way, paving the way for future event-related implementation on neuromorphic chips.

## B Definition of Event Density

In our experiments, we investigated the relationship between the location of the split point determined by the SNN and the density of the corresponding event stream. The event density, denoted as $D(t)$, measures the concentration of events in a given event stream over time. It is mathematically defined as the rate at which events occur per timestep, expressed as a function of:

$$D(t) = \frac{\delta N}{\delta t} \tag{11}$$

where $D(t)$ is the event density at timestep $t$, $\delta N$ is the number of events occurring within a small time interval $\delta t$ around $t$. This definition enables a precise quantification of event concentration, offering insights into the temporal distribution of events at any given moment. Our empirical analysis shows that there is a significant correlation between the split points of the SNN and the event density. Notably, in regions of higher event density, the SNN exhibits a tendency to perform more frequent split. In contrast, in regions with lower event density, the number of split is lower. This behavior emphasizes the adaptability of our proposed dynamic slicing method, SpikeSlicer, to the fluctuating information density in the event stream.

# C  Sensitivity Analysis of Fixed Event Slicing Method

To demonstrate that events are sensitive to slicing by fixed methods, and to emphasize the importance of proposing a dynamic event slicing approach, we have conducted a total of 60 experiments with different models to investigate the impact of different slicing techniques and different numbers of slices on the performance in downstream tasks.

In our experiment, we employed two fixed slicing methods: (1). *Slicing with a fixed number of events*, and (2). *Slicing with a fixed duration*. $N$ denotes the number of resulting event slices. Experimental results are detailed as follows in Tab. 8 and Fig. 7.

Table 8: The sensitivity analysis of fixed event slicing on N-Caltech101. The results demonstrate that the event is sensitive to the fixed slicing method (slicing by fixed time or event count), thereby affirming the need for proposing a dynamic slicing method.

| N-Caltech101 | $N$ | 2 | 4 | 6 | 8 | 10 | 12 | 14 | 16 | 18 | 20 | 22 | 24 | 26 | 28 | 30 | Mean | Var |
|---|---|---|---|---|---|---|---|---|---|---|---|---|---|---|---|---|---|---|
| ResNet18 | *Fixed Count* | 70.96 | 75.26 | 75.39 | 75.30 | 76.09 | 73.95 | 74.09 | 73.80 | 76.40 | 75.39 | 75.45 | 73.60 | 71.94 | 71.01 | 71.17 | **73.98** | **3.33** |
| ResNet18 | *Fixed Time* | 62.90 | 72.64 | 76.38 | 74.48 | 74.91 | 73.70 | 74.30 | 74.69 | 76.95 | 74.75 | 74.46 | 74.42 | 71.61 | 71.52 | 69.69 | **73.16** | **10.80** |
| ResNet34 | *Fixed Count* | 72.19 | 75.55 | 76.98 | 78.22 | 77.14 | 77.40 | 76.78 | 76.90 | 78.14 | 77.06 | 76.91 | 74.85 | 74.76 | 76.91 | 73.07 | **76.19** | **2.90** |
| ResNet34 | *Fixed Time* | 65.42 | 75.92 | 78.29 | 78.20 | 78.48 | 76.22 | 77.76 | 76.57 | 75.94 | 76.80 | 76.61 | 75.91 | 75.11 | 74.76 | 74.19 | **75.74** | **9.15** |

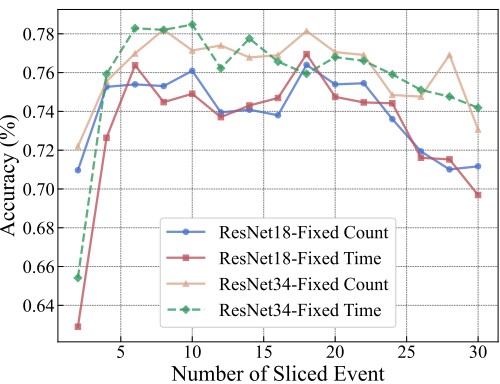

Figure 7: Visualization of sensitivity analysis on N-Caltech101 dataset. The fluctuations in accuracy for different numbers of sliced event with different fixed slicing methods are significant, demonstrating that events are very sensitive to fixed slicing methods.

The results indicate significant fluctuations (large variance) in downstream performance based on the slicing method and the number of slices used. We believe this addition effectively demonstrates the sensitivity of event streams to slicing techniques, **confirming the need for our motivation to propose dynamic slicing of event streams.** Additionally, the accuracy achieved using the dynamic slicing method (82.54% by ResNet34) surpasses that of any fixed slicing approach (with the highest being 78.48%), further substantiating the efficacy of the dynamic method in our study.

# D  Illustration of SpikeSlicer vs. Fixed Slicing Method

In order to more intuitively show the difference between our dynamic event slicing method and the traditional fixed slicing method, we specifically illustrate these methods through Fig. 1.

Fig. 1(a) denotes that each resulting sliced event has the same duration, and Fig. 1(b) denotes that the number of event points contained in each resulting sliced event is the same. Since event stream are usually unevenly distributed, a fixed cutting method often leads to non-uniform event information (*e.g.*, in scenarios with changing motion speed). In contrast, our approach decides the optimal slice position through feedback from downstream ANN by using an SNN as the event slicing trigger.

# E  Difference between Event Slicing and Event Representation

It is worth noting that **our work focuses on the slicing of the event stream rather than focusing on event representation**. Event representation refers to the process of event information extraction

Table 9: Experiments on different event representations with fixed (including fixed time and fixed event count) or dynamic slicing methods. Our SpikeSlicer yields significant improvement when using different event representation methods.

| DVSGesture | Event Frame | Event Spike Tensor | Voxel Grid |
|---|---|---|---|
| Fix Duration | 93.75% | 93.75% | 88.54% |
| Fix Event Count | 93.06% | 94.79% | 88.19% |
| **SpikeSlicer (ours)** | **94.79%** | **95.49%** | **89.24%** |

that is performed after the event stream has been sliced into sub-event stream, and the resulting event representation meets the neural network input requirements. Thus, our dynamic slicing process and event representation can be used at the same time, either better slicing or representation method benefits the feature extraction with neural network, thus improving performance.

To validate the effectiveness of our slicing approach, we supplement the event-based recognition task below. We compare the downstream performance of three different event representation methods (including Event Frame [10], Event Spike Tensor (EST [9]) and Voxel Grid [12]) on the DVSGesture dataset under fixed slicing and dynamic slicing method in Tab. 9.

## F   Definition of Raw Event Group

Based on the definition of event field from [9], we here define a general version of raw event group representation as a mapping $\mathcal{M} : \mathcal{E} \mapsto \mathcal{T}$ between the set $\mathcal{E}$ and a tensor $\mathcal{T}$:

**Definition 1** (Raw event group). *Based on a measurement* $\mathbb{1}_{\text{condition}}$, *raw event group are grid-like tensors defined in continuous space and time:*

$$G_\pm(x, y, t, \text{condition}) = \sum_{e_k \in \mathcal{E}_\pm} \mathbb{1}_{\text{condition}}(x, y, t)\delta(x - x_k, y - y_k)\delta(t - t_k), \qquad (12)$$

where $\pm$ denotes the event polarity; $\mathbb{1}_{\text{condition}}$ sets the specific approach of representation to each event, *e.g.*, condition $= \{\sum_{e_k} = M\}$ denotes the number of event points in each grid-like tensor is fixed to $M$, *i.e.*, slicing event stream $\mathcal{E}$ by *event count*; or condition $= \{\Delta t = t_x\}$ denotes the time interval of event points in each grid-like tensor is fixed to $t_x$. $G_\pm$ are grid tensors with $x \in \{0, 1, ..., W - 1\}, y \in \{0, 1, ..., H - 1\}$, and $t \in \{t_0, t_0 + \Delta t_{x1}, ..., t_0 + B\Delta t_{xB}\}$, where $t_0$ is the first time stamp, $\Delta t_{xi}$ is the bin size determined by the splitting condition, and $B$ is the number of temporal bins. Eq. (12) converts raw event into grid-like raw event group by a Dirac pulse [9] in the space-time manifold. The resulting $G_\pm$ gives a continuous time representation of $\mathcal{E}$ which preserves the event's information.

However, if such raw event groups are then converted into event representation (*e.g.*voxel grid [22], time surface [43]), the generated event representation is imprecise due to the fact that the process of $\mathbb{1}_{\text{condition}}$ is fixed, leading to both spatial and temporal information loss. The main objective of this study is to solve the problem of fixed slicing of the event stream and to provide a dynamic segmentation scheme.

## G   Reason for Using No-reset Membrane Potential

We first recall the definition of original membrane potential $V[n]$:

$$V[n] = \beta V[n-1] + \gamma I[n], \qquad (13)$$
$$S[n] = \Theta(V[n] - \vartheta_{\text{th}}), \qquad (14)$$
$$V[n] = V[n](1 - S[n]) + V_{\text{reset}}, \qquad (15)$$

where the the neuron will reset its membrane potential to $V_{\text{reset}} < \vartheta_{\text{th}}$ at time $n$ once it trigger a spike $S[n]$. As described in Sec. 3.3, we choose to guide the membrane potential without reset stage (Eq. 15) $U[n]$ (or named no-reset membrane potential), instead of the normal membrane potential $V[n]$. The no-reset membrane potential is defined similarly as $V[n]$:

$$U[n] = \beta U[n-1] + \gamma I[n], \qquad (16)$$
$$S[n] = \Theta(U[n] - \vartheta_{\text{th}}), \qquad (17)$$

but the neuron does not reset its membrane potential in this condition. The reason behind this choice is that the reset process will affect the guidance of the $V[n]$. Specifically, suppose we expect the neuron to fire a spike at $n + 1$, but if a spike just occurs at time $n$, the membrane potential $V[n]$ will reset to $V_{\text{reset}}$, consequently leading to a small value for $V[n + 1]$. Based on the SPA-Loss function, $V[n + 1]$ would then be guided by a large expected membrane potential value (above the threshold). However, this would incorrectly guide the membrane potential after resetting to the desired membrane potential, rather than guiding the true membrane potential as intended. Therefore, we choose to use the no-reset membrane potential $U[n]$ to effectively guide the spiking neuron to fire spike at the specified location.

## H    Proof of Proposition

**Proposition 1.** *Suppose the input event cell sequence has length $N$, desired spiking time is $n^*$ ($n^* \in \{0, 1, ..., N\}$), the membrane potential at time $n^*$ satisfying the constraints:*

$$V_{th} \leq U[n^*] \leq \max(\beta V_{th} + \gamma I[n^*], V_{th}), \tag{18}$$

*where $I[n^*]$ is the input synaptic current from Eq.1. Then the spiking neuron fires a spike at time $n^*$ and does not excite spikes at neighboring moments.*

*Proof.* Here we consider two conditions that affect the spiking state at moment $n^*$:

(1)  Membrane potential at time $n^*$ is too small to emit a spike.

(2)  Membrane potential at time $n^*$ is too large, affecting neighboring moment spiking states.

To satisfy the condition (1), we only need to guide the membrane potential $U[n^*]$ to reach the threshold $V_{th}$ at time $n^*$, thus the upper bound of $U[n^*] = V_{th}$. In condition (2), we need to consider the state of the membrane potential at $n^* - 1$ and $n^* + 1$. We first exhibit the accumulation rules of membrane potential:

$$U[n^*] = \beta U[n^* - 1] + \gamma I[n^*] \tag{19}$$

However, if $U[n^*]$ is too large, this may cause the membrane potential $U[n^* - 1]$ to exceed the threshold and occur spike generation prematurely, and then the membrane potential will immediately drop to a reset value (Eq. 15). This will leave the membrane potential $U[n^*]$ at a very low value, making it difficult to trigger a spike. Hence, we should control the membrane potential not to exceed the threshold value at moment $n^* - 1$:

$$U[n^* - 1] = \frac{(U[n^*] - \gamma I[n^*])}{\beta} \leq V_{th} \tag{20}$$

$$\Rightarrow \quad U[n^*] \leq \beta V_{th} + \gamma I[n^*] \tag{21}$$

Thus, the upper bound of $U[n^*] = \beta V_{th} + \gamma I[n^*]$. However, if the leaky factor $\beta$ is small, there exists a possibility that $\beta V_{th} + \gamma I[n^*] \leq V_{th}$, thus we set the upper bound of $U[n^*]$ as $\max(\beta V_{th} + \gamma I[n^*], V_{th})$.

Next, we consider whether the spike at time $n^*$ affects the pulse state at time $n^* + 1$ Since the neuron at the $n^*$ moment has already completed the spike generation before accumulating $U[n^* + 1]$. Therefore the membrane potential at $n^*$ does not affect the neuronal state at $n^* + 1$. In sum, if the membrane potential satisfies: $V_{th} \leq U[n^*] \leq \max(\beta V_{th} + \gamma I[n^*], V_{th})$, the spiking neuron fires a spike at moment $n^*$ and does not excite spikes at neighboring moments.

## I    More Explanations in Spiking Position-aware Loss

### I.1    Details in Membrane Potential-driven Loss

In Sec. 3.3.1, we explore the range of the expected membrane potential $U[n^*]$ and ensure its rationality by proposition 1 (proved in Appendix. H). To understand the setting of the membrane potential more easily, we visualize the boundary cases in Fig. 8.

The lower bound case means that the membrane potential at the desired index should be at least $V_{th}$ to activate a spike, and the upper bound case guides the membrane potential to exceed the threshold but prevents generating spikes in the previous time step. Hence, the desired membrane potential should be bounded in $[V_{th}, \max(\beta V_{th} + \gamma I[n^*], V_{th})]$ and $\alpha \in [0, 1]$ (Eq. 8) balances the desired membrane potential $U[n^*]$ between $U_{\text{lower}}$ and $U_{\text{upper}}$.

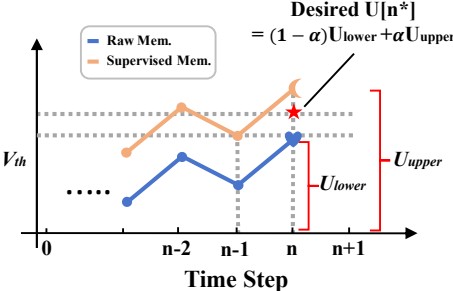

Figure 8: Visualization of the boundary cases when controlling the desired membrane potential, where the 'heart-like' point denotes the lower bound case and the 'moon-like' point denotes the upper bound case.

## I.2 Details in Linear-assuming loss

As described in Sec. 3.3.2, suppose we expect the SNN to trigger a spike at a later time, if there exists a hill effect, the earlier membrane potential always reaches the excited state sooner and turns the neuron into the resting state to suppress the spiking generation at later moments. To address this challenge, we expect the later membrane potential to satisfy: (1) the later membrane potential should be larger than the current membrane potential to reverse the hill effect, and (2) the later membrane potential should exceed the threshold to fire a spike. Hence, we assume that the membrane potential in this condition should increase monotonically with the time step, as illustrated in Fig. 9.

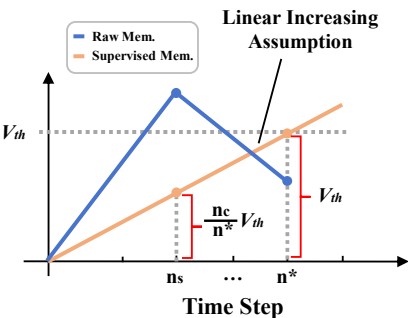

Figure 9: Visualization of our expected linearly increasing membrane potential.

Then we use the LA-Loss to supervise the membrane potential at $n_c$ to reach $\frac{n_c}{n^*} V_{th}$ in order for the latter membrane potential at the $n^*$ to reach $V_{th}$, satisfying both (1) and (2).

## I.3 Details of Dynamic Hyperparameter Tuning

To deeply explore the control of the hyperparameter $\alpha$, we first analyze the effect of $\alpha$ in Eq. 8. When $\alpha$ reaches its maximum value (*i.e.* $\alpha = 1$), the desired membrane potential evolves to $U_{upper}$, which corresponds to a situation where the membrane potential just reaches the threshold at the previous moment (in Fig. 8). That is, as $\alpha$ increases, the previous moment is more likely to generate a spike, driving the spiking time earlier. Recalling observation 2, taking a large-$\alpha$ scenario as an example, the SNN fails to spike at a later time step due to the alpha being too large, limiting the neuron's ability to spike later. Hence, we expect the $\alpha$ to decrease to allow the desired index to decrease as well; similarly for small $\alpha$ scenarios. To summarize, we hope the $\alpha$ is updated in the same direction as the desired index, we update $\alpha$ by setting:

$$\alpha.\text{grad} = ||\sum_{i}^{N_s}(n^{*i} - n_c^i)/N_s||_2^{2'}, \tag{22}$$

$$\alpha \leftarrow \alpha - 2 \cdot \eta \sum_{i}^{N_s}(n^{*i} - n_c^i)/N_s, \tag{23}$$

where grad denotes the gradient of $\alpha$ and $'$ denotes derivative operation. The explanations of other math symbols can be found in the main text and Alg. 2.

## J    Visualization of the SNN Training Process

To validate whether the proposed feedback-update strategy can serve a guiding role during the initial stages of training, we have visualized the training process of the SNN and presented it in Figure 1 of the supplementary PDF rebuttal file. As anticipated, the training of the SNN exhibited fluctuations during the initial training stage, which might be attributed to the instability of the event quality obtained from dynamic slicing at this early phase. However, as training progressed, the loss of the SNN gradually stabilized and decreased, converging towards a desired outcome. Correspondingly, the slicing times progressively converged towards the desired spiking index. Therefore, although initial exploration may require several steps, our proposed training method is capable of offering effective guidance.

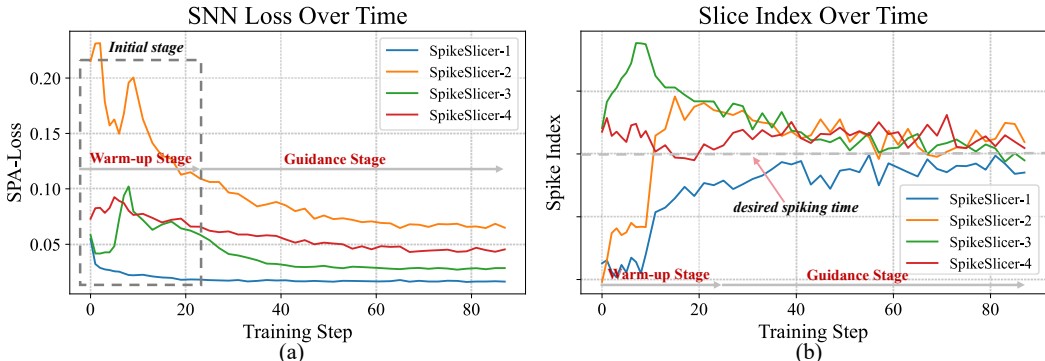

Figure 10: **Visualization of the SNN Training Process.** As shown in the gray box in (a), the SNN loss fluctuates considerably during the initial training stage. This is due to the instability in the quality of dynamic sliced events during the early phase (warm-up stage). However, as training progresses, the SNN loss gradually stabilizes and converges (guidance stage), with the corresponding slicing time (b) also converging towards the desired spike index.

## K    Implementation Details in Toy Experiments

To validate the effectiveness of SPA-Loss, we set up the toy task:

*Input $N$ randomized event cells, expect the SNN to slice at a specified time step $n^*$ and there exists a certain probability of interfering with SNN to slice at other time steps.*

In detail, we set the time step within the range $[1, 30]$ and the max number of iterations as 800. We utilize a lightweight convolutional SNN with random initialization. We compare our proposed SPA-Loss function with common mean square error (MSE-Loss) and cross-entropy loss (CE-Loss). The results in Fig. 4 demonstrate that our proposed SPA-Loss can guide the SNN to spike at the desired timestep accurately with fewer convergence iterations than standard MSE-Loss and CE-Loss, paving the way for complex event-based vision tasks in Sec. 4.2. More experiments on the beginner's arena are shown in Appendix N.

## L    Implementation Details in Event-based Task

We adopt a spiking neural network with structure: {*16C3-GN-IF-AvgP2-32C3-GN-IF-AvgP2-64C3-GN-IF-AdaP2-LN-IF-LN-IF*}, which consists of three convolutional layers and two linear layers, no residual block or attention are used. *{i}C{j}* denotes a convolutional layer with the output channel $i$ and the kernel size $j$; *GN* denotes group normalization; *AvgP{k}* and *AdaP{k}* mean the average pooling and adaptive pooling with kernel size $k$; *LN* denotes the linear layer. We choose the IF neuron as the activation function. We adopt the SGD optimizer and set the initial learning rate as 1e-4, along with the cosine learning rate scheduler. SNN models are trained with batch size 32. Each experiment is conducted in an NVIDIA 4090 GPU.

## L.1 Event-based Object Tracking

**Datasets.** The FE108 dataset [3] is an extensive event-based dataset for single object tracking, including 21 different object classes and several challenging scenes, *e.g.*, low-light (LL) and high dynamic range (HDR). The event streams are captured by a DAVIS346 event-based camera, which equips a 346x260 pixels dynamic vision sensor (DVS). We choose 54 sequences for training ANNs, 22 sequences for training SNNs and the rest 32 sequences for testing.

**Evaluation Metrics.** To show the quantitative performance of each tracker, we utilize three widely used metrics: success rate (Suc.), precision rate (Prec), normed precision rate (N-Prec), and overlap precision (OP). These metrics represent the percentage of three particular types of frames. Success rate is the frame of that overlap between the ground truth and the predicted bounding box is larger than a threshold; Precision rate focuses on the frame of the center distance between ground truth and predicted bounding box within a given threshold; $OP_{thres}$ represents SR with $thres$ as the threshold. We employ the area under curve (AUC) to represent the success rate. The precision score is associated with a 20-pixel threshold.

**Label Settings.** In this paper, since the original event dataset only provided labels at fixed frame rates, we employed a linear interpolation method to obtain corresponding labels for each more refined event cell. For example, suppose that in the original event dataset, a sub-event stream $E$ with a period of $T$ (i.e., $t \in [t_1, t_2]$) has labels $\{l_{t_1}, l_{t_2}\}$ corresponding to moments $t_1, t_2$, respectively. If the number of event cells in this interval is $N$, then each event cell represents the event with the time range of $\{[t_1, t_1 + \frac{T}{N}], [t_1 + \frac{T}{N}, t_1 + 2\frac{T}{N}], ...\}$, and the label for each interval can be derived through linear interpolation using $\{\tilde{l}_{t_1}, l_{t_2}\}$ and the number of event cells $N$. Thus, predictions at any slicing interval have corresponding labels for supervised learning. To ensure fairness, all tracking experiments in this paper utilize the aforementioned method to process the event dataset.

## L.2 Event-based Recognition

**DVS-Gesture.** The DVS-Gesture [34] dataset contains 11 hand gestures from 29 subjects under 3 illumination conditions, recorded by a DVS128.

**N-Caltech101.** The N-Caltech101 dataset [35] incorporates 8,831 event-based images, with a 180×240 resolution and 101 classes, generated from the original Caltech101 dataset through an event-based sensor.

**DVS-CIFAR10.** The DVS-CIFAR10 dataset [36] is an event-stream dataset designed for object classification. It consists of 10,000 event streams, created by converting the frame-based images from the CIFAR-10 dataset using an event-based sensor with a resolution of 128×128 pixels. This dataset presents an intermediate level of difficulty, featuring 10 distinct classes.

**SL-Animals.** The SL-Animal database [37] features DVS recordings of individuals performing sign language gestures representing various animals, captured as a continuous spike flow with very low latency. This dataset includes approximately 1100 samples from 58 subjects, each performing 19 different sign language gestures in isolation across various scenarios, offering a challenging evaluation platform for this emerging technology.

## M Theoretical Energy Consumption Calculation

To calculate the theoretical energy consumption, we begin by determining the synaptic operations (SOPs). The SOPs for each block in the SNN can be calculated using the following equation:

$$\text{SOPs}(l) = fr \times T \times \text{FLOPs}(l) \tag{24}$$

where $l$ denotes the block number in the SNN, $fr$ is the firing rate of the input spike train of the block and $T$ is the time step of the spike neuron. $\text{FLOPs}(l)$ refers to floating point operations of $l$ block, which is the number of multiply-and-accumulate (MAC) operations. And SOPs are the number of spike-based accumulate (AC) operations.

To estimate the theoretical energy consumption of SNN, we assume that the MAC and AC operations are implemented on a $45nm$ hardware, with energy costs of $E_{MAC} = 4.6pJ$ and $E_{AC} = 0.9pJ$, respectively. According to [44, 45], the calculation for the theoretical energy consumption of SNN is

given by:

$$E_{\text{Diffusion}} = E_{MAC} \times \text{FLOP}^1_{\text{SNN}_{\text{Conv}}}$$
$$+ E_{AC} \times \left( \sum_{n=2}^{N} \text{SOP}^n_{\text{SNN}_{\text{Conv}}} + \sum_{m=1}^{M} \text{SOP}^m_{\text{SNN}_{\text{FC}}} \right) \tag{25}$$

where $N$ and $M$ represent the total number of layers of Conv and FC, $E_{MAC}$ and $E_{AC}$ represent the energy cost of MAC and AC operation, $\text{FLOP}_{\text{SNN}_{\text{Conv}}}$ denotes the FLOPs of the first Conv layer, $\text{SOP}_{\text{SNN}_{\text{Conv}}}$ and $\text{SOP}_{\text{SNN}_{\text{FC}}}$ are the SOPs of $n^{th}$ Conv and $m^{th}$ FC layer, respectively.

## N  More Experiments on Beginner's Arena

**Problem Setup.** To verify the accuracy of our proposed slicing method, we expect SNN can slice events at the specified time step. We set up two scenarios and only show the difficult task (II) in the main text:

- *Task (I): Input $T$ identical event cells, expect the SNN to slice at a specified time step $T^*$.*

- *Task (II): Input $T$ randomized event cells, expect the SNN to slice at a specified time step $T^*$, but there exists a certain probability of interfering with SNN to slice at other locations.*

Task (I) aims to verify whether our proposed slicing strategy can accurately locate the optimal point; To test the robustness of our method, task (II) simulates complex event stream processing with random inputs and adds random noise to affect the SNN with wrong labels.

Table 10: Results on simple event slicing tasks with SPA-Loss.

|  | **Input Size** | **Time Steps** | **Parameter** | **Iterations to Convergence ↓** |
|---|---|---|---|---|
| *Task (I)* | $32 \times 32$ | 30 | 0.52M | 75 |
| | $64 \times 64$ | 30 | 2.02M | 81 |
| *Task (II)* | $32 \times 32$ | 100 | 0.52M | 29 |
| | $64 \times 64$ | 100 | 2.02M | 88 |

We adopt a lightweight SNN (0.25M/2.02M) for our experiments. $T^*$ is randomly selected within range $[0, T]$. The experimental results presented are the average of the results obtained by setting up three random seeds. As shown in Tab. 10, SNN requires only a small number of iterations to converge to the specified slicing time based on the SPA-Loss. For the complex task with random inputs and disturbances in task (II), SNN can still converge fast and even faster to find the specified cut point compared with task(I). *This simple experiment demonstrates that SPA-Loss can effectively supervise the SNN pulsing at the specified location, which paves the way for experiments on adaptive event slicing in real scenarios.*

## O  Statistics of Dynamic Slicing (SpikeSlicer) vs. Fixed Slicing

In this section, we compare the statistics results of the resulting events sliced by different slicing methods.

*Symbol Description: the total event stream $E$; the resulting sliced sub-event stream list by SNN: $E_{beef} = [E_1^b, ..E_{N_1}^b]$; the resulting sliced sub-event stream list by fixed slicing method: $E_{fixtime} = [E_1^f, ..E_{M_1}^f]$.*

In the tracking task, the average duration of each sub-event stream $E_k^b (k \in [1, N_1])$ is 65ms (corresponding to 13 event cells, and the duration of the event stream contained in each event cell is 5ms). The maximum duration of each sub-event stream is 100ms, and the minimum duration is 30ms, while for our comparison of the slicing-by-fixed-time approach, the duration of each sub-event stream $E_j^f (j \in [1, M_1])$ is fixed at 75ms. The following Tab. O and Fig. 11 show the specific statistics of adaptive slicing vs. fixed slicing:

## P  Statistics of Resulting Slicing Duration

To further verify the stability and effectiveness of the dynamic slicing method, we explore the results of our method by changing the number of event cells in the event recognition task. $N_{cell}$ indicates

| Method | Avg Cell Num | Var Cell Num | Avg Duration | Min Duration | 25th Duration | 75th Duration | Max Duration |
|---|---|---|---|---|---|---|---|
| SpikeSlicer | 12.99 | 3.96 | $\sim 65$ms | 25ms | 50ms | 80ms | 100ms |
| Slice by fixed duration | 15 | 0 | 75ms | // | // | // | // |

Table 11: Statistic results of dynamic slicing method (our SpikeSlicer) and fixed slicing method.

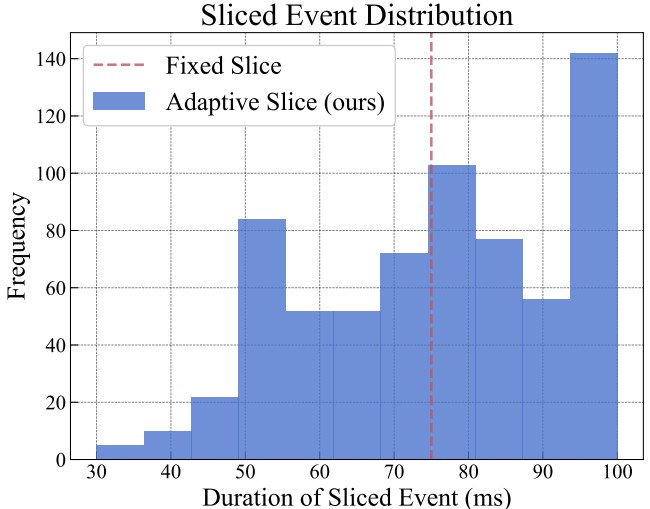

Figure 11: Visualization of sliced event distribution. Our method can return the sliced events with different durations, while the fixed method can only generate sliced events with fixed durations.

that an event stream is divided into $N_{cell}$ event cells, and the larger the $N_{cell}$ implies that the event stream is divided into more fine-grained event cell sequences that are capable of better represent the raw event stream (as mentioned in Sec. 3.1).

*Calculation Process: Suppose the whole event stream (duration = $T$) is divided into 15 event cells, if the SNN is trained to sliced the event with 2.42 (average) event cells, which means that the sliced sub-event stream $E_k^b$ contains event data which lasts a duration of $\frac{1}{15} * 2.42 * T = 16.13\%T$.*

Table 12: Experiments of SpikeSlicer with different event cell numbers $N$. The resulting sliced event group always has a similar time interval in various $N$ conditions.

| $N_{cell}$ | **15** | **20** | **25** |
|---|---|---|---|
| Avg Cell Num | 2.42 | 3.15 | 4.77 |
| Percentage of Duration | 16.13% | 15.75% | 19.08% |

The experimental results show that the percentage of the duration of each sub-event stream to the total event stream duration after the adaptive slicing is relatively stable, *i.e.*, the fineness of the event cell does not affect the event information contained in each sub-event stream after the slicing process, which proves the robustness and effectiveness of the dynamic slicing process of our method. We also illustrate the event percentage change during training in Fig. 12.

## Q   More Experiments with Latest Models

To enhance the credibility and robustness of our results, we have incorporated state-of-the-art models: Swin Transformer (SwinT [46]) and Vision Transformer (ViT [47]), to further validate the efficacy of our algorithm in event-based recognition tasks (Tab. 13):

*Experiment Settings: We choose SwinT-small and ViT-small for comparisons on the DVSGesture dataset. Other settings are consistent with the main experiments.*

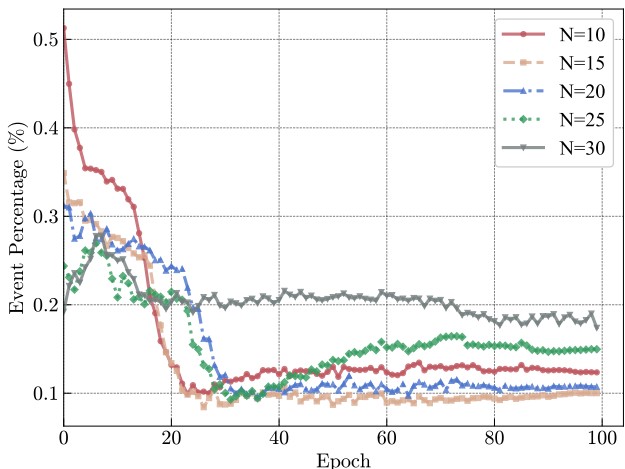

Figure 12: Visualization of the percentage of event in various $N_{cell}$ conditions during training. SpikeSlicer can always split and obtain sub-event groups with similar time intervals.

Table 13: Experiments of utilizing SpikeSlicer on latest recognition backbones.

| Method | Random Slice | Fixed Slice | Ours |
|--------|--------------|-------------|------|
| SwinT [46] | 88.19 | 89.93 | **91.67(+1.74%)** |
| ViT [47] | 87.50 | 85.07 | **88.54(+3.47%)** |

Results demonstrate that the SpikeSlicer also yields performance improvement in recognition tasks with the latest backbones.

## R   More Experiments with Complex Neuromorphic Dataset

To further validate the proposed approach, we conduct experiments on more complex neuromorphic N-ImageNet [48]:

Table 14: Experiments of utilizing SpikeSlicer on complex dataset N-ImageNet [48]

| Slice Method | Random Slice | Fixed Slice | Ours |
|--------------|--------------|-------------|------|
| ResNet-18 | 40.98 | 39.43 | **45.48(+6.05%)** |

## S   Impact Statement

This paper proposes an effective event processing method and also provides a novel SNN-ANN cooperation paradigm, aiming to inspire further research and development in energy-efficient and high-performance computing. We do not anticipate a direct negative impact from our work.

## T   Summary

To sum up, SpikeSlicer is designed as a **plug-and-play** algorithm for dynamic event stream slicing. It is benchmarked against baselines that employ fixed event stream slicing methods, proving the effectiveness of our method. In addition, our approach is versatile and can be applied in any event-based vision task, not limited to recognition or single object tracking scenarios.

Notably, SpikeSlicer also provides a brand-new **SNN-ANN cooperation paradigm**, where the SNN acts as an efficient, low-energy data processor to assist the ANN in improving downstream performance, injecting new perspectives and potential avenues of exploration.

