# OpenReview forum: "Spiking Neural Network as Adaptive Event Stream Slicer"
_NeurIPS.cc/2024/Conference — NeurIPS 2024 poster_

### Official Review · Reviewer_cS27 · 2024-06-18

**Soundness:** 3
**Presentation:** 3
**Contribution:** 3
**Rating:** 6
**Confidence:** 5

**Summary:**

This paper proposes to use spiking neural networks (SNNs) to slice the event stream in an adaptive manner before passing the voxelized events to the downstream inference model. The first step of the proposed method divides the input event stream into voxelized event cells with the same temporal interval. An SNN, constructed to have a scalar output, then takes the event cells recurrently. The timestamps when the SNN generates spikes are considered to be the slicing positions. The slicing SNN and the downstream inference model are trained together using the membrane potential-driven loss and the linear-assuming loss. The feedback-update strategy allows the two networks to be trained end-to-end. Extensive experiments on a toy example, object tracking, and recognition demonstrate that the proposed method can be easily integrated into existing models, bringing a noticeable performance improvement.

**Strengths:**

1. The paper focuses on an interesting yet underexplored problem, which is to use a data-driven model to adaptively construct the event voxels. The proposed method is intuitive, and the key idea is convincing. It is clear that the authors have put in great effort in preparing this submission.

2. The key technical contribution involves two parts. First, the feedback-update strategy allows supervision signals from the downstream ANN to back-propagate to the SNN. Additionally, the membrane potential-driven loss and the linear-assuming loss control the spiking time through the supervision of the membrane potential value. The two parts complement each other, leading to an end-to-end trainable model.

3. Additionally, the paper also discusses how the hyperparameter $\alpha$ can be tuned together with the SNN weights and analyzes the implication of different $\alpha$ values to the spiking behavior.

4. The experiments are very extensive. The SpikeSlicer has been validated on several event-based applications, demonstrating its prediction quality, efficiency, and the fact that it can be easily incorporated into existing models.

**Weaknesses:**

1. Despite the strengths above, the key design appears to be a bit simple. As a potential NeurIPS paper, this work is relatively weak on the technical sophistication and theoretical insights. However, this is complemented by extensive experimental evaluation and empirical analysis.

2. While SNNs are efficient and consume less energy than ANNs, SNNs are also less capable than ANNs. Since the speed of the entire SNN+ANN prediction pipeline is going to be slow anyway, it may be worthwhile to investigate whether using an ANN as an event slicer can lead to better prediction quality.

3. While the proposed losses are justified by proposition 1 and empirical analysis, it is unclear if the proposed feedback-update strategy is the best way to identify the desired trigger time $n^*$. In particular, it is unclear if the argmin operator can return any meaningful signal during the initial training stages.

**Questions:**

I like the key idea and hope the paper can be accepted.

I encourage the authors to respond to the third bullet point in the "Weaknesses" section. Additional experiments addressing the second bullet point are **not** expected as part of the rebuttal.

**Limitations:**

Yes, the authors have adequately addressed the limitations and potential negative societal impact of their work.

---

> ### Author Rebuttal · Authors · 2024-08-07
>
> **Q1:** *While SNNs are efficient and consume less energy than ANNs, SNNs are also less capable than ANNs. Since the speed of the entire SNN+ANN prediction pipeline is going to be slow anyway, it may be worthwhile to investigate whether using an ANN as an event slicer can lead to better prediction quality.*
>
> **A1:**
> Thank you for your suggestion! Firstly, it is indeed true that the current capabilities of SNNs are not on par with those of ANNs, and simulation speeds on GPUs can be slower for SNNs. However, our motivation for utilizing SNNs as a slicer is twofold: (1). SNNs are low-energy consuming, and (2). They have the potential for extremely high operational efficiency and speed when deployed on neuromorphic chips, a fact that has been substantiated by several studies [1,2,3]. Therefore, this paper takes this motivation as a starting point and proposes an event slicer based on SNNs. We fully acknowledge and appreciate your point that an ANN-based event slicer could potentially achieve higher performance. Given that the forward propagation in ANNs does not involve the concept of binary signals, the slicing process would require further design and consideration, which indeed warrants future investigation. Thank you again for your valuable suggestion!
>
>
>
>
> ---
>
>
> **Q2:** *While the proposed losses are justified by proposition 1 and empirical analysis, it is unclear if the proposed feedback-update strategy is the best way to identify the desired trigger time $n^*$. In particular, it is unclear if the argmin operator can return any meaningful signal during the initial training stages.*
>
> **A2:**
> Thank you for your question! To validate whether the proposed feedback-update strategy can serve a guiding role during the initial stages of training, we have visualized the training process of the SNN and presented it in **Figure 1** of the supplementary PDF rebuttal file. As anticipated, the training of the SNN exhibited fluctuations during the initial training stage, which might be attributed to the instability of the event quality obtained from dynamic slicing at this early phase. However, as training progressed, the loss of the SNN gradually stabilized and decreased, converging towards a desired outcome. Correspondingly, the slicing times progressively converged towards the desired spiking index. Therefore, although initial exploration may require several steps, our proposed training method is capable of offering effective guidance. We will also attempt further optimizations to enhance its efficiency. Thank you for your comment！
>
>
>
> ---
>
>
> ***Reference:***
>
> [1] Roy A, Nagaraj M, Liyanagedera C M, et al. Live demonstration: Real-time event-based speed detection using spiking neural networks. CVPR, 2023.
>
> [2] Yu F, Wu Y, Ma S, et al. Brain-inspired multimodal hybrid neural network for robot place recognition. Science Robotics, 2023.
>
> [3] Viale A, Marchisio A, Martina M, et al. Lanesnns: Spiking neural networks for lane detection on the loihi neuromorphic processor. IROS, 2022.

---

> > ### Comment · Reviewer_cS27 · 2024-08-09
> > **Response to the Rebuttal**
> >
> > Dear Authors,
> >
> > Thank you for submitting the rebuttal! Given the unanimous support, this paper is likely going to be accepted. This is a solid work that deserves acceptance. However, I encourage the authors to investigate using ANNs to predict the slicing positions in the future.
> >
> > Sincerely,
> > Reviewer cS27

---

> > > ### Author Response · Authors · 2024-08-09
> > > **Response to Reviewer cS27**
> > >
> > > Dear reviewer cS27,
> > >
> > > Thank you very much for supporting and recognizing our work!! We will explore the topic of the ANN-based event slicer in the future. Thanks!
> > >
> > > Best,
> > > authors

---

> > ### Comment · Reviewer_HdTR · 2024-08-14
> >
> > Thanks for the clarifications and effort. I'm going to keep my rating. I don't have any more questions.

---

> > > ### Author Response · Authors · 2024-08-14
> > > **Response to Reviewer HdTR**
> > >
> > > Dear reviewer HdTR,
> > >
> > > Thank you very much for your support!! We will incorporate the supplemented results into the future manuscript. Thanks!
> > >
> > > Best, authors

---

### Official Review · Reviewer_HdTR · 2024-07-13

**Soundness:** 2
**Presentation:** 3
**Contribution:** 2
**Rating:** 6
**Confidence:** 3

**Summary:**

This work proposes a novel method for adaptively sample event data and subsequently preprocess it, utilizing a spiking neural networks (SNNs) as module.

The sampling method involves a feedback mechanism that triggers the activation of the SNN.

**Strengths:**

Tests are done on dataset with different lighting conditions. Being robust to different event rates.

**Weaknesses:**

The experiments conducted do not contain tasks such as optical flow, object detection, or image reconstruction. The type of tasks tested is limited.

**Questions:**

no questions

**Limitations:**

The implementation of the code appears to be challenging, which may affect its reproducibility.
The application of this algorithm in embedded systems seems to be constrained due to the use of Spiking Neural Networks (SNNs).
Furthermore, it is unclear whether the code will be made publicly available.

---

> ### Author Rebuttal · Authors · 2024-08-07
>
> **Q1:** *The experiments conducted do not contain tasks such as optical flow, object detection, or image reconstruction. The type of tasks tested is limited.*
>
> **A1:**
> Thank you for your suggestion. Due to limited time and resources, we have endeavored to incorporate a variety of task types. Specifically, we have added **four** tasks: (1). Lips Reading. (2). Human Gait Recognition. (3). Camera Pose Relocalization. (4). Object Detection in Real-world Environment. Details of all experimental setups and results are as follows:
>
>
> ### **(1) Lips Reading**
>
> *Introduction and Experiment Settings:* Lips reading seeks to decipher textual content via the visual cues provided by the speaker's lip movements. We opt the DVS-Lip dataset [1]and crop the event into 88x88. Other experiment settings are aligned with those of the main experiment. The performance is measured by the accuracy.
>
>
> | **Method** | Fixed | **Ours** |
> |:-------|:-------|:------:|
> | ResNet-18 | 16.23 | **18.50(+2.17%)**|
> | ResNet-34 | 18.14 |**19.08(+0.94%)**|
>
> ### **(2) Human Gait Recognition**
>
> *Introduction and Experiment Settings:* Human gait recognition aims to determine human identities based on their walking patterns captured by the sensors. We select the EV-Gait dataset [2] and the input resolution is 128x128.
>
>
> | **Method** | Fixed | **Ours** |
> |:-------|:-------|:------:|
> | ResNet-18 | 88.40 | **89.15(+0.75%)**|
> | ResNet-34 | 84.90 |**89.10(+4.20%)**|
>
>
>
> ### **(3) Camera Pose Relocalization**
>
>
> *Experiment Settings:* Camera pose relocalization aims to train several scene-specific neural networks to accurately relocalize the camera pose. We choose the IJRR dataset [3] and select the standard SP-LSTM [4] as our baseline. The input resolution is set as 180x240. The performance is measured by the distance error and the degree error. ME and AE denote the median and average error, respectively. We abbreviate meter and degree as m and e.
>
>
> ||Shape Rotation||||Shape Translation||||
> |------------------------|--------------------|---------------------|---------------------|----------------------|------------------------|--------------------|---------------------|---------------------|
> | **Method**                 | **ME-m** | **ME-d** | **AE-m** | **AE-d** |**ME-m** | **ME-d** | **AE-m** | **AE-d** |
> | VGG-LSTM               | 0.065             | 26.014              | 0.081               | 38.398               | 0.177 | 8.623	| 0.185	| 13.176 |
> | VGG-LSTM **(+SpikeSlicer)**| **0.055**             | **21.292**              | **0.072**              | **22.666**               | **0.173**	|**7.161**|	**0.181**	|**10.276**|
>
>
> ### **(4) Object Detection in Real-world Environment**
>
>
> *Collection Settings:* To validate the effectiveness of our method in real-world scenarios, **we collected a small-scale, multi-object, event-based detection dataset under low-exposure conditions in an indoor setting.** We adopted a DAVIS-346 event-based camera with resolution 260x346 and recorded the dataset in the indoor under-exposure environment. Two volunteers participated in the recording and performed random body movements during the recording. The training set and testing set contains 125 and 18 event streams, respectively.
>
> *Experiment Settings:* We select three models as our baselines: YOLO-Tiny, YOLO-Nano and YOLO2. We crop the event as 320x320. We use the most commonly used mean average precision as our evaluation metric. We report the $\mathbf{AP}$ of the person category.
>
>
>
>
> |   |  | Fix Slice|  |  | SpikeSlicer |  |
> |-----------|--------------|-------------|--------------|---------------|--------------|---------------|
> | **Method**    | $AP_{0.25}$  | $AP_{0.5}$  | $AP_{0.75}$  | $AP_{0.25}$ | $AP_{0.5}$  | $AP_{0.75}$ |
> | Yolo-tiny     | 48.46                | 46.80               | 38.24                | **54.58** **(+6.12)**    | **49.57** **(+2.77)**   | **44.87** **(+6.63)**   |
> | Yolo-nano     | 59.85                | 55.17               | 23.24                | **60.56** **(+0.71)**    | **55.33** **(+0.16)**   | **32.66** **(+9.42)**    |
> | Yolo-v2       | 62.21                | 59.65               | 46.45                | **62.62** **(+0.41)**    | **62.04** **(+2.39)**   | **48.38** **(+1.93)**    |
>
>
>
> ### **(5) Summary**
> Based on the experimental results provided, we have further confirmed the efficacy of SpikeSlicer across various tasks. Additionally, SpikeSlicer has demonstrated positive outcomes in real-world scene detection. We will incorporate these results into future manuscript.
>
>
>
>
>
> ---
>
>
>
> **Q2:** *The implementation of the code appears to be challenging, which may affect its reproducibility. The application of this algorithm in embedded systems seems to be constrained due to the use of Spiking Neural Networks (SNNs). Furthermore, it is unclear whether the code will be made publicly available.*
>
> **A2:** Thanks!
>
> *(1) Reproducibility:* We have provided detailed experimental settings in the appendix, including the network structure and algorithm details. We plan to open-source our code in the future and greatly appreciate your suggestion.
>
> *(2) Applicability of SNNs in Embedded Systems:* Indeed, numerous works have successfully deployed SNN algorithms on hardware devices, achieving high accuracy, remarkable operational efficiency, and low power consumption. For instance, the work by Yu et al. [5] published in Science Robotics, deployed SNN on the neuromorphic chip Tianjic [6], combined with an event-camera mounted on a quadruped robot to achieve efficient and low-energy locomotion in complex environments. Similarly, Roy et al. [7] deployed SNN on the Loihi chip to realize real-time event-based detection. There are also many other fields, e.g., EEG processing [8], robotics [9] , etc., where the deployment of SNN has been successfully implemented. Therefore, the application of SNNs in embedded systems is feasible.
>
> ---
>
> Due to word limitations, the reference part is moved to the official comment.

---

> > ### Comment · Reviewer_HdTR · 2024-08-14
> >
> > I have no further questions and will keep my rating. Thank you to the authors for the clarifications.

---

### Official Review · Reviewer_ij7X · 2024-07-13

**Soundness:** 3
**Presentation:** 3
**Contribution:** 3
**Rating:** 5
**Confidence:** 4

**Summary:**

The authors designed a plug-and-play event processing method, SpikeSlicer, to split event streams with an adaptive amount. The proposed method is a lightweight SNN, constrained by a custom Spiking Position-aware Loss (SPA-Loss) to regulate neuron states. Additionally, a downstream ANN refines the slicing decisions using a feedback-update training strategy.

**Strengths:**

* The proposed plug-and-play event processing method, SpikeSlicer, based on SNN representation, which can be used for various vision tasks with event cameras.
* The downstream ANN refines the slicing decisions using a feedback-update training strategy, and the performance of the downstream ANN provides feedback to adjust the representation.
* Experimental results demonstrate that SpikeSlicer can effectively enhance the performance of object tracking and recognition with event cameras, while also leveraging the advantages of neural computation in processing speed and power consumption.

**Weaknesses:**

* The comparison algorithm for event-based object tracking, DiMP, is from 2019. Why not try the latest methods? In recent years, many studies have focused on improving the effectiveness of event stream representation to enhance the performance of event vision tasks.
* Many methods for object detection and tracking with event cameras have not been compared.

**Questions:**

The input to SNNs also involves a time constant definition, which implicitly includes the concept of a time window. How did the authors determine this time constant?

**Limitations:**

There is not much discussion on this aspect.

---

> ### Author Rebuttal · Authors · 2024-08-07
>
> **Q1:** *The comparison algorithm for event-based object tracking, DiMP, is from 2019. Why not try the latest methods? In recent years, many studies have focused on improving the effectiveness of event stream representation to enhance the performance of event vision tasks; Many methods for object detection and tracking with event cameras have not been compared.*
>
> **A1:**
> Thank you for your suggestion! Due to time and resource constraints, we have incorporated a recent tracking model, TaMOs (2024) [1], and tested it on the FE108 low light dataset. The results are displayed in the table below, from which it can be observed that the tracking performances under the SpikeSlicer gain improvement. Consequently, this substantiates the efficacy of our method.
>
>
> |       |                  |                     |                 |                  |
> |---------------|:------:|:----:|:----:|:----:|
> |   **Methods**            | RSR  | OP$_{0.50}$ |  RPR  | Norm RPR  |
> | TaMOs | 46.35 | 30.53          | 87.75	          | 62.59  |
> | TaMOs **(+SpikeSlicer)** | **46.56** | **31.20**         |  86.17 |	**65.54** |
>
> ---
>
> **Q2:** *Many methods for object detection and tracking with event cameras have not been compared.*
>
> **A2:**
> Thank you for your comment! Regarding the experiments on object detection, we collected a multi-object event-based dataset and tested it using multiple baselines. The experimental results are presented in the rebuttal PDF document, with the specific experimental setups outlined in the response to reviewer HdTR. These results demonstrate the performance improvements in real-world object detection scenarios.
> We will include these results in a future version of our manuscript. Thanks!
>
>
>
> ---
>
> **Q3:** *The input to SNNs also involves a time constant definition, which implicitly includes the concept of a time window. How did the authors determine this time constant?*
>
> **A3:**
> Thank you for your question! The input of the SNN is $N$ event cells, so the duration of each event cell (time window) is given by $\delta t = T/N$. We do not set the time window directly; instead, we specify the number $N$, and the time window is implicitly determined during the generation of the event cells.
>
>
> ---
>
> ***Reference:***
>
> [1] Mayer C, Danelljan M, Yang M H, et al. Beyond SOT: Tracking Multiple Generic Objects at Once. WACV, 2024.

---

> > ### Comment · Reviewer_ij7X · 2024-08-14
> >
> > Thank you for your response and for providing a comparison with the WACV2024 method. However, I think the authors still lack comparisons with recent algorithms focusing on enhancing event representation, as this is a common challenge in event camera applications. Additionally, I suggest referring to the event-based vision resources on GitHub (https://github.com/uzh-rpg/event-based_vision_resources?tab=readme-ov-file#feature-detection-and-tracking), which continuously compile works on detection and tracking algorithms.
> >
> > Regarding point A3, if the authors use a fixed number of event cells 𝑁 as input to the model, does that mean the time interval 𝛿𝑡 becomes a variable? Based on the operating principle of DVS, the faster the motion, the more events are generated in a short period, and conversely, fewer events are generated during slower motion. Therefore, I find the authors' claim of "adaptive" somewhat ambiguous. How does this differ from the method illustrated in Fig. 1(b)?
> >
> > Considering these concerns, I may need to reconsider the reproducibility of this work as a plug-and-play module.

---

> > > ### Author Response · Authors · 2024-08-14
> > > **Response to Reviewer ij7X**
> > >
> > > **Q4:** *I think the authors still lack comparisons with recent algorithms focusing on enhancing event representation...*
> > >
> > > **A4:** Thank you for your question! **Our approach focuses on slicing the event stream rather than proposing a new event representation method** (refer to Line 27-32 of the original manuscript). Our method is designed to be compatible with **any** event representation. In terms of comparing different event representation methods, we have already provided comparisons in Table 4 of the manuscript, which includes EST, Timesurface, and voxel grid methods. These comparisons demonstrate the effectiveness of our approach across various event representations.
> > >
> > > As you mentioned, DVS cameras can produce uneven events due to variations in motion speed. **Our motivation is precisely to address the challenge of performance degradation caused by traditional methods that apply fixed slice to the event stream, which is independent of the age of the model.**
> > >
> > > We have validated the effectiveness of our method through extensive experiments across a range of models and settings, including tracking, action recognition, and object recognition tasks. Additionally, following your suggestion, we have included a 2024 model within the short rebuttal period. Moreover, **in the rebuttal period**, we have supplemented with **four** new downstream tasks: lip reading, gait recognition, pose relocalization, and object detection. The summary of these experiments (**more than 100**) are summarized in the table below:
> > >
> > > | | **Main Experiments**  |  |  | **Ablation**| **Rebuttal Supplementary** | | | |
> > > |:-------:|:-------:|:-------:|:------:|:------:| :------:|:------:|:------:|:------:|
> > > | Name | Object Tracking | Action Recognition | Object Recognition| Event Representation| Lips Reading| Gait Recognition| Pose Estimation | Object Detection|
> > > | Experiment settings | 4 backbones, 4 scenes, 3 slicing methods | 3 backbones, 2 datasets, 3 slicing methods |3 backbones, 2 datasets, 3 slicing methods | 2 tasks, 3 representation methods, 3 slicing methods | 2 backbones, 1 datasets, 2 slicing methods | 2 backbones, 1 dataset, 2 slicing methods|1 backbone, 2 datasets, 2 slicing methods | 1 self-collected dataset, 3 backbones, 2 slicing methods |
> > > |Experiment Number | 48  |  18 | 18 |18 | 4 | 4 | 4| 6|
> > >
> > > Our extensive experiments can support our method's effectiveness.
> > >
> > > Thank you for sharing the information on the latest models. As the deadline is approaching, we will try our best to include the latest models in future versions. We sincerely appreciate your suggestion!
> > >
> > > ---
> > >
> > > **Q5:** *...How does this differ from the method illustrated in Fig. 1(b)?*
> > >
> > > **A5:** I think there might be some misunderstandings here.
> > >
> > > First, the "number of event cells" and the "event count" in Figure 1(b) represent two **different** concepts.
> > > Allow me to clarify the dynamic slicing process of the SNN as follows:
> > >
> > > Suppose we have an event stream from the FE108 airplane dataset with a total duration of 24994 $\mu s$, and the SNN's inputs are 10 event cells. Each event cell (as defined in Definition 1) corresponds to an event voxel with a duration of $\delta t = \frac{24994}{10} \mu s \approx 2.5$ ms . These 10 event cells are then fed into the SNN (based on an Integrate-and-Fire neuron without decay). If a spike occurs at the 3-rd position, we extract the sub-event stream within the time interval of [0, 3*2.5] = [0, 7.5] ms. At this stage, any event representation method can be used to convert the sub-event stream into an event representation, which can then be fed into downstream tasks. Since the SNN's spike output is dynamic, the event slicing process is dynamic as well.
> > >
> > > We hope our response helps to clarify your concerns. Thank you for your questions!

---

> > > > ### Comment · Reviewer_ij7X · 2024-08-14
> > > >
> > > > Thank the authors for their swift reply. I have no further questions and hope the authors will consider providing open access to the data and code.

---

> > > > > ### Author Response · Authors · 2024-08-14
> > > > > **Response to Reviewer ij7X**
> > > > >
> > > > > Dear Reviewer ij7X,
> > > > >
> > > > > Thank you very much for your support!! We will consider providing our codes.
> > > > >
> > > > > Best,
> > > > > authors

---

### Author Response · Authors · 2024-08-07
**Reference part of the Response to Reviewer HdTR**

Due to word limitations, the reference part of the response to reviewer HdTR is moved to the official comment, thanks!

***Reference:***

[1] Tan G, Wang Y, Han H, et al. Multi-grained spatio-temporal features perceived network for event-based lip-reading. CVPR, 2022.

[2] Wang Y, Du B, Shen Y, et al. EV-gait: Event-based robust gait recognition using dynamic vision sensors. CVPR, 2019.

[3] Mueggler E, Rebecq H, Gallego G, et al. The event-camera dataset and simulator: Event-based data for pose estimation, visual odometry, and SLAM. IJRR, 2017.

[4] Nguyen A, Do T T, Caldwell D G, et al. Real-time pose estimation for event cameras with stacked spatial lstm networks. CVPRW, 2017.

[5] Yu F, Wu Y, Ma S, et al. Brain-inspired multimodal hybrid neural network for robot place recognition. Science Robotics, 2023.

[6] Deng L, Wang G, Li G, et al. Tianjic: A unified and scalable chip bridging spike-based and continuous neural computation. IEEE Journal of Solid-State Circuits, 2020.

[7] Roy A, Nagaraj M, Liyanagedera C M, et al. Live demonstration: Real-time event-based speed detection using spiking neural networks. CVPR, 2023.

[8] Kumar N, Tang G, Yoo R, et al. Decoding eeg with spiking neural networks on neuromorphic hardware. TMLR, 2022.

[9] Viale A, Marchisio A, Martina M, et al. Lanesnns: Spiking neural networks for lane detection on the loihi neuromorphic processor. IROS, 2022.

---

### Author Rebuttal · Authors · 2024-08-07

We sincerely thank the reviewers for their thoughtful comments and feedback. We appreciate that all reviewers agreed that the idea of using spiking neural network (SNN) for dynamic event slicing is interesting and evaluated the paper with positive scores. Below, we address the primary concerns raised by the reviewers and summarize the revision:


- Reviewer ij7X suggested to provide comparison with the latest method. In response, we supplemented a recent tracking model and demonstrated the effectiveness of our method.
- HdTR suggested implementing a wider variety of tasks for testing. In response, we tried our best to incorporate more task types and supplemented **four** tasks. In particular, we conducted Lips Reading, Human Gait Recognition, Camera Pose Relocalization and Object Detection in **Real-world** Environment.
- Reviewer C27 raised a good question, inquiring whether the proposed feedback-update strategy could serve a guiding role during the initial stages of training. In response, we visualized the training process of the SpikeSlicer, as depicted in Figure 1 of the submitted PDF file. By analyzing the changes in the training curves, we demonstrated the effectiveness of our proposed strategy.
- For real-world detection, we **collected a multi-object, event-based detection dataset** under low-exposure conditions.
- We **submitted a rebuttal PDF** file which includes all the experiment results.
- We visualized examples from the newly added downstream task datasets, which are presented in Figure 2.
- We showcase the experimental setup for collecting real-world event dataset and visualized some samples, as shown in Figure 3.
- The experimental results for the supplementary tracking and downstream tasks (lip reading, gait recognition, camera pose relocalization, and detection) are compiled in Figure 2-3 and Table 1-4 for easy reference.


Thanks for your time and effort in reviewing our paper, your suggestions have greatly helped to enhance the article!

---

### Decision · Program_Chairs · 2024-09-25

**Decision:**

Accept (poster)

**Comment:**

The authors present an approach for splitting event streams adaptively. The authors argue that typically, when dealing with input event streams, and when these events are processed by an algorithm that operates using frame-based input, an approach for slicing the input events is needed. This is usually done by splitting at uniform time intervals, or by splitting when the total spike density exceeds some threshold. The authors propose a system where a spiking neural network is actually responsible for adaptively deciding when to do the splitting, based on a feedback mechanism utilizing feedback provided by the ANN that does the top-level detection/classification task.

A concern I have with this work is that it assumes there is always a need to convert event streams into groups of spikes, or frames. This is only needed when you want to provide a frame-based representation as input to a more traditional neural network that operates based on the concept of a frame. When dealing with neuromorphic hardware and algorithms, you need to tone-down the concept of a frame that is associated with more traditional neural networks, in order to achieve efficiency.

Nevertheless, all reviewers seem to believe that the algorithm is useful when the event data is meant to be consumed by more traditional ANN algorithms that consume frame-like inputs. The reviewers indicate that the paper does a thorough experimental comparison and seems to demonstrate improved accuracy.

The authors indicated in the rebuttal that they plan to release source code in the "future". This was in response to a concern expressed by one of the reviewers regarding reproducibility. After reading the paper I tend to agree with this reviewer comment. To ensure reproducibility of this work I would like to ask the authors that they release their source code, so that it is feasible for someone to reproduce their results. I believe this would also increase the impact of their work.

In summary there seems to be consensus among the reviewers, and therefore I recommend acceptance. As I indicated above, and as a condition of acceptance, I would like the authors to also release source code with the paper's publication (as they indicated in their rebuttal that they would do in the future) in order to make this work easily reproducible.